# In Silico and Cellular Differences Related to the Cell Division Process between the A and B Races of the Colonial Microalga *Botryococcus braunii*

**DOI:** 10.3390/biom11101463

**Published:** 2021-10-05

**Authors:** Xochitl Morales-de la Cruz, Alejandra Mandujano-Chávez, Daniel R. Browne, Timothy P. Devarenne, Lino Sánchez-Segura, Mercedes G. López, Edmundo Lozoya-Gloria

**Affiliations:** 1Genetic Engineering Department, CINVESTAV-IPN Irapuato Unit, Irapuato 36824, Mexico; xochitl.morales@cinvestav.mx (X.M.-d.l.C.); lino.sanchez@cinvestav.mx (L.S.-S.); 2Instituto Tecnológico y de Estudios Superiores de Monterrey, Campus Irapuato, Irapuato 36670, Mexico; amandujano@tec.mx; 3Department of Biochemistry and Biophysics, Texas A&M University, College Station, TX 77843, USA; dbrowne.up@gmail.com (D.R.B.); tpd8@tamu.edu (T.P.D.); 4Pacific Biosciences, Chicago, IL 60606, USA; 5Biochemistry and Biotechnology Department, CINVESTAV-IPN Irapuato Unit, Irapuato 36824, Mexico; mercedes.lopez@cinvestav.mx

**Keywords:** *Botryococcus braunii*, chlorophyll, cyclin-dependent kinases, multinucleated, retinoblastoma

## Abstract

*Botryococcus braunii* produce liquid hydrocarbons able to be processed into combustion engine fuels. Depending on the growing conditions, the cell doubling time can be up to 6 days or more, which is a slow growth rate in comparison with other microalgae. Few studies have analyzed the cell cycle of *B. braunii*. We did a bioinformatic comparison between the protein sequences for retinoblastoma and cyclin-dependent kinases from the A (Yamanaka) and B (Showa) races, with those sequences from other algae and *Arabidopsis thaliana*. Differences in the number of cyclin-dependent kinases and potential retinoblastoma phosphorylation sites between the A and B races were found. Some cyclin-dependent kinases from both races seemed to be phylogenetically more similar to *A. thaliana* than to other microalgae. Microscopic observations were done using several staining procedures. Race A colonies, but not race B, showed some multinucleated cells without chlorophyll. An active mitochondrial net was detected in those multinucleated cells, as well as being defined in polyphosphate bodies. These observations suggest differences in the cell division processes between the A and B races of *B. braunii*.

## 1. Introduction

The colonial microalgae *Botryococcus braunii* produces large amounts of useful hydrocarbons and value-added compounds like pigments, lipids, polysaccharides, and other biopolymers [1]. These colonies consist of individual cells embedded within an extracellular matrix (ECM) in which liquid hydrocarbons are stored [2]. According to the hydrocarbons produced and molecular phylogeny, *B. braunii* is classified into A, B, and L races [3]. Race A synthesizes fatty acid derived C_23_–C_33_ odd-carbon-numbered alkadienes and/or alkatrienes [4]. The Yamanaka strain of the A race used here produces only C_25_–C_31_ alkadienes [5]. Race B synthesizes the C_30_–C_37_ triterpenoid-based hydrocarbons called botryococcenes [6], and the L race synthesizes a C_40_ tetraterpenoid hydrocarbon known as lycopadiene [7].

According to the ITIS Report (Integrated Taxonomic Information System) [8], *Botryococcus braunii* has the Taxonomic Serial No. 6308, and the taxonomic hierarchy is as follows: Kingdom Plantae, Subkingdom Viridiplantae, Infrakingdom and Division Chlorophyta, Class Trebouxiophyceae, Order Trebouxiales, and Family Botryococcaceae. A phylogenetic analysis of two isolates of *B. braunii* race A, one from race B, and one from race L, using the small nuclear sequence of the 18S rRNA, was performed [9]. Phylogenetic trees were constructed with distances of maximum parsimony (MP) and maximum likelihood (ML), and their reliability was calculated using the bootstrap and bayesian methods. The analysis showed that the four *B. braunii* isolates form a monophyletic group that approximates the genus Choriscystis of the class Trebouxiophyceae. Weiss et al. [10] generated a phylogeny of a *B. braunii* race B using the 18S rRNA sequence with the ML method, which confirmed that it belonged to the Trebouxiophyceae class. Then, Kawachi et al. [11] carried out a 18S rRNA phylogenetic study to relate the hydrocarbons produced by *B. braunii* with the molecular phylogeny of the microalgae. A phylogenetic tree using 31 axenic strains of the microalgae isolated from various regions of Japan was generated and gas chromatography−mass spectrometry was included. The results showed that all of the analyzed strains belonged to the Trebouxiophyceae class. This analysis revealed the existence of three major clades (I, II, and III) within this class. Clade I consisted mainly of race A strains, clade II of race B strains, and clade III of a mixture of races L and S strains. Clades II and III were divided into two subclades (II1, II2, III1, and III2). Subclade III2 was made up of six race S strains and subclade III1 was separated from the previous one because three race L strains were grouped in it [11].

*Botryococcus braunii* race B (Showa) is a colonial green microalga present in brackish and fresh water. It has been found all around the world, and produces long-chain hydrocarbons, which reduce the density of large masses, allowing them to float on the water surface. Each colony is composed of 50 to 100 pear-shaped non-flagellated cells held together in a hydrocarbon extracellular matrix [2]. The diversity of 10 strains of *B. braunii* race A isolated in different aquatic environments was studied regarding their morphology, hydrocarbon, and fatty acid profiles [12]. Irregular green, lax spherical small clusters of pyriform cells (14–15 μm), forming colonies of 21 μm to 70 μm in size, were observed. The size and number of lipid bodies were related to the cell cycle stage. Both races A and B are autotrophic and mixotrophic [13]. Noguchi et al. [14] describes the presence of autospores as two daughter cells into a mother cell wall, involved in the cell division mechanism in a colony of non-specified race of *B. braunii*. Weiss et. al. [2] proposed that *B. braunii* colonies grow to a specific size, and are then fragmented to produce smaller colonies, which grow again. Suzuki et al. [15] reported that during the growth stage of *B. braunii* race B (Showa), each cell divides into two daughter cells, but no information was given about the growth of the colonies. Single cells of *B. braunii* race B (Showa) released with glycerol were unable to survive unless they were in a high concentration (>2 × 10^7^ cells/mL) [16].

The cell doubling time of *B. braunii* is low in comparison with other fast dividing species of microalgae [17]. The cell doubling time of *Chlamydomonas reinhardtii* is between 7–15 h [18,19,20,21], for *Ostreococcus tauri*, it is between 11–20 h [22,23], for *B. braunii* Showa strain it is between 1.3–6.0 days, and for *B. braunii* A races it is between 2.3–10.6 days [24]. Unfortunately, there is very little information about the cell cycle of *B. braunii*. Most studies involving cell division or the cell cycle have indirectly studied these processes while analyzing cell biology or hydrocarbon production. One of these reports was done in an electron microscopy study of the *trans*-Golgi network (TGN) structure using an unspecified race of *B. braunii* [14]. The study described the formation of two autospores within the mother cell after division in a binary fission mode. Another study on the synthesis of hydrocarbons during the cell cycle of the A race was done after the synchronization of cells in the S phase using 5-aminouracil [25]. In that work, a pair of autospores in the mother cell was reported 21 h after 5-aminouracil removal. Suzuki et al. [15] used the B race of *B. braunii* to analyze cell ultrastructure by electron microscopy. They induced cell division with illumination and provided a general description of the mitosis of *B. braunii* [15]. Thus, it is assumed that race A and B cells divide by binary fission.

A plant cell cycle master regulator is the retinoblastoma (RBR) tumor suppressor, and phosphorylation−dephosphorylation reactions of this protein regulate the RBR pathway [26,27]. The RBR pathway involves the E2F-family of transcription factors [28], as well as inhibitors and activators of cyclin-dependent kinases (CDKs) [29]. Briefly, RBR binds the transcription factor E2F, preventing the start of the cell cycle from proceeding from the G1 to S phase. Once RBR is phosphorylated by specific CDKs, the phosphorylated RBR releases E2F, which starts the cell cycle. All of these elements play critical roles in the regulation of cell cycle progression and cell death of plants [30,31]. In green algae, when E2F is phosphorylated and other transcription factors are available, cell division begins. This happens at the committed point (CP), when the G1/S-phase transition takes place [32]. The role of the RBR pathway is so important that some authors suggested that the multicellularity evolution did not rely upon the de novo evolution of genes, but the evolution of undifferentiated colonies correlates with RBR cell cycle regulatory pathway evolution [33]. In resting animal cells, it was reported that low levels of RB phosphorylation were present, but in rapidly proliferating cells the RB protein was highly phosphorylated. Maximal RB phosphorylation was associated with the S phase of the cell cycle [26]. In addition, RB was found to maintain the chromosome structure and mediate protein degradation in mechanisms that are transcription independent [34].

CDKs also play a key role in cell cycle regulation. After a comparison of almost 50 plant CDK-related cDNAs with their animal and yeast counterparts, it was concluded that common cell cycle tools may be used in different organisms and in various combinations to achieve cell cycle regulation [35]. The CDK enzymes have a conserved PSTAIRE motif that is essential for cyclin binding [36], and these enzymes are involved in the success of reaching the CP and mitosis [37,38]. Maximum activities of CDKA1 and CDKB1 were observed at CP in *C. reinhardtii* [39]. It is known that CDKC;2 modulates both cell division and the drought response in *Arabidopsis* [40]. It is accepted that CDKCs control cellular differentiation through RBR inactivation. A CDC2-like kinase (CDC2L5) with a PITAVRE motif, which is a variation of the previously mentioned PSTAIRE motif, was reported in sea urchin (*Sphaerechinus granularis*), *Drosophila*, and *Caenorhabditis elegans*. Proteins related to pre-mRNA splicing mechanisms have sequence similarities with these enzymes, which suggests a certain role in the regulation of the gene expression [41]. CDKD1 of *O. tauri* and CDKD;1 and CDKD;3 of *A. thaliana* are involved in the control of mitosis [42,43]. The CDKEs enzymes of all of the analyzed organisms have the same conserved SPTAIRE motif, and they control cell expansion and cell fate specification [44]. CDKF;1 is also part of the kinase network like CDKDs, and it is present in *A. thaliana* but does not have a very conserved motif [45]. CDKG;1 and CDKG;2 of *A. thaliana* are involved in a thermo-sensitive mRNA splicing cascade [46]. CDKH1 from *C. reinhardtii* has a PVSTIRE motif and forms a sister group with the CDKCs family [39]. CDKI1 controls the flagellar length and flagellar assembly in *C. reinhardtii* and has a PDVVVRE motif [47,48].

A critical cell size is required for attaining the CP and for starting the reproductive sequence. Several models have been proposed to explain the connection between cell size and cell cycle progression, and all models conclude that RBR and CDKs play critical roles in sensing the cell size [49]. The growth rate and cell size of algae are regulated by the cell cycle, and two types of cell division have been described: binary fission, which is the most common, and multiple fission, which has evolved in some species of algae or is present under specific growth conditions [49]. Binary fission is the mechanism by which one cell divides into two daughter cells, while in multiple fission one cell gives rise to more than two cells [50].

It has been reported that in algae, cell cycle progression depends on the accumulation of storage compounds such as starch, lipids, and polyphosphates (poly-P) [49]. Poly-P contribute to energy generation through ATP production and cell metabolism, and induces cell differentiation via gene induction. The number of daughter cells and the progress of the cell cycle is regulated by growth conditions such as temperature, light, and the amount of nutrients and reserves [49,50]. For many years, these factors have been used to synchronize the cell cycle within a population of cells in order to carry out accurate genetic studies [51].

In the current work, we used the amino acid sequences of RBR and CDKs from several microalgae to identify homologous proteins of the A and B races of *B. braunii* for comparative studies. Additionally, we noticed differences in cell division between the races by analyzing the number of cell nuclei through microscopy. Fluorescence microscopy of the A and B races was used to identify and localize the nuclei and chloroplasts, as well as the mitochondrial net and poly-P bodies. Surprisingly, we found differences in the number of nuclei, the presence of chlorophyll, the kind of mitochondrial net, and poly-P between *B. braunii* race A and race B, which suggest a difference in the respective cell cycles and cell division processes.

## 2. Materials and Methods

### 2.1. Strains and Culture Conditions

*Botryococcus braunii*, of race A Yamanaka strain [5] and race B Showa or Berkeley strain [52], cultures were grown in modified Chu 13 medium (900 mL) at pH 7.4 [53]. To diminish the culture media evaporation, cultures were grown under aerated conditions by supplying hydrated filter-sterilized ambient air generated by passing the air through a sterile water flask. Cultures were grown at 25 °C with fluorescent lighting giving an intensity of 50 μmol m^−2^ s^−1^ of photons using 59 W fluorescent bulbs (Philips F96T8 59W/850 Single Pin ALTO Plus; Philips Mexico Commercial, Mexico City, Mexico) under a 12/12 h light/dark photoperiod. Samples were taken at different times for microscopic observations. The growth of both races was followed spectrophotometrically, starting with an initial inoculum of 0.03 average at optical density (OD) 680 nm. The stationary phase was assessed when OD 680 nm reached a maximum and constant value along the time, as mentioned before for each race. The OD 680 nm at the stationary phase was 0.216 ± 0.028 at 24 days for race A, and 0.270 ± 0.05 at 30 days for race B. Race A grew little faster than race B.

### 2.2. Bioinformatic Identification of Retinoblastoma (RBR) and Cyclin-Dependent Kinases (CDKs)

The protein sequences of the RBR protein were retrieved from the National Centre for Biotechnology Information (NCBI) database [54] for the following species: *Chlamydomonas reinhardtii* (MAT3/RBR (XP_001696629.1)), *Ostreococcus tauri* (MAT3/RBR (OUS45688.1)), *Gonium pectorale* (MAT3/RBR (BAN18532.1)), *Volvox carteri f nagariensis* female (f) (MAT3/RBR (ABM47317.1)), *Volvox carteri f nagariensis* male (m) (MAT3/RBR (ADI46925.1)), and *Arabidopsis thaliana* (RBR1 (NP_566417.3)). The CDK protein sequences were retrieved from the NCBI database [54], for the following species: *C. reinhardtii* (CDKA1 (XP_001698637.1), CDKB1 (XP_001701299.1), CDKC1 (XP_001694199.1), CDKD1 (XP_001694537.1), CDKE1 (PNW83964.1), CDKG1 (XP_001696492.1), CDKG2 (XP_001701126.1), CDKH1 (XP_001702056.1), and CDKI1 (XP_001700559.1)); *O. tauri* (CDKA1 (XP_003078530.1), CDKB1 (XP_003083211.1), CDKC1 (AAV68597.1), CDKD1 (AAV68598.1), CDKE1 (XP_022840036.1), CDKG1 (XP_003074327.1), and CDKG2/CDK10 (XP_003080520.2)); *G. pectorale* (CDKA1 (KXZ46110.1), CDKB1 (KXZ43845.1), CDKC1 (KXZ43562.1), CDKD1 (KXZ54250.1), CDKE1 (KXZ45461.1), CDKG1 (KXZ52035.1), CDKH1 (KXZ53820.1), and CDKI1 (KXZ49794.1)); *V. carteri* f*. nagariensis* (CDKA1 (XP_002949867.1), CDKB1 (XP_002947156.1), CDKC1 (XP_002954450.1), CDKD1 (XP_002954735.1), CDKE1 (XP_002957533.1), CDKG1 (XP_002946192.1), CDKH1 (XP_002956993.1), and CDKI1 (XP_002956880.1)); and *A. thaliana* (CDKA;1 (NP_566911.1), CDKB1;1 (NP_190986.1), CDKB1;2 (NP_001031507.1), CDKB2;1 (NP_177780.1), CDKB2;2 (NP_173517.1), CDKC;1 (NP_196589.1), CDKC;2 (NP_201301.1), CDKD;1 (NP_177510.1), CDKD;2 (NP_176847.1), CDKD;3 (NP_173244.1), CDKE;1 (NP_201166.1), CDKF;1 (NP_001329562.1), CDKG;1 (OAO92015.1), and CDKG;2 (NP_001154456.1)).

The above RBR and CDK sequences were used to identify the corresponding sequences in the A and B races of *B. braunii* using Basic Local Alignment Search Tool (BLAST). For the A race, the RBR and CDK sequences were identified from an unpublished transcriptome developed by the Devarenne lab, which was processed with TransDecoder v3.0.1 to identify the coding regions and to predict the proteins from the transcripts—this resulted in 20,644 open reading frames (ORFs). B race sequences were identified using the B race genome [55], housed at Phytozome [56], the NCBI [57], and the Joint Genomic Institute [58]. The RBR and CDK sequences were identified using BLASTp [59], with the previously mentioned query sequences from *Ostreococcus tauri*, *Chlamydomonas reinhardtii*, *Gonium pectorale*, *Volvox carteri* f. *nagariensis*, and *Arabidopsis thaliana*. The CDK sequences were determined based on the percent identity of the queries and the conservation of the cyclin-binding domain; the protein kinase domain; and the A, B, and DUF3452 domain for the RBR sequence [60]. A search for domains within the retrieved sequences was performed in the Batch CD-Search program [61], with an expected value of 0.01 against the CDD-55570 PSSMs database [62].

This process identified a single RBR protein from the B race, but did not identify an RBR from the A race. Thus, the B race amino acid sequence was used to identify the A race RBR protein from the A race transcriptome. For the CDK sequences, this process identified 28 annotated CDK-encoding sequences in the B race and after screening these sequences for the PSTAIRE motif, we selected eight CDKs to use for the analysis. To identify the CDKs in race A, the selected race B CDK protein sequences were used to construct a multiple-sequence alignment with Clustal Omega. Using a hidden Markov model (HMM)-based approach [63], this alignment was converted to a profile HMM using HMMER v3.1b2 [64]. Then, the profile HMM was screened against the predicted complete race A protein set and 526 total matches were identified. The top 31 best matches (2.5 × 10^−71^ to 5.6 × 10^−60^ E score) were selected as CDKs candidates in race A (Appendix A). As was done for the B race, the A race CDK sequences were screened for the presence of the PSTAIRE motif, and nine sequences were selected (Appendix A).

### 2.3. Identification of Potential Phosphorylation Sites in the RBR Protein Sequence

To identify the potential CDKs-phosphorylation sites in the RBR protein sequence, the NetPhos 3.1 Server [65] was used [66,67]. The best predictions were generated and a consensus phosphorylation site, where the minimum required sequence was S/T-P, and CDKs specific sites (S/T-P-X-K/R where X is any residue) were subsequently identified [68,69,70,71] (Appendix A).

### 2.4. Phylogenetic Analysis

Using the identified A and B race RBR and CDK amino acid sequences, an evolutionary phylogenetic analysis was performed in comparison with those sequences from the Chlorophyta algae *O. tauri*, *C. reinhardtii*, *G. pectorale*, and *V. carteri* f. *nagariensis*, as well as *A. thaliana*. The evolutionary history was inferred using the Maximum Likelihood method and Poisson correction model [72]. The bootstrap consensus tree inferred from 1000 replicates was taken to represent the evolutionary history of the analyzed taxa [73]. Branches corresponding to partitions reproduced in less than 50% bootstrap replicates were collapsed. Initial tree(s) for the heuristic search were obtained automatically by applying Neighbor-Joining and BioNJ algorithms to a matrix of pairwise distances estimated using the Poisson model, and then selecting the topology with superior log likelihood value. This analysis involved 63 amino acid sequences. All positions containing gaps and missing data were eliminated (complete deletion option). There was a total of 195 positions in the final dataset. Evolutionary analyses were conducted in MEGA X [74,75].

### 2.5. Synchronization of B. braunii Cultures

We followed a previously described procedure [25]. Briefly, a sterile stock solution of 2.5 mM 5-aminouracil was added to the A and B race cultures into a modified Chu 13 medium at pH 7.4. After one-month of growth with 5-aminouracil, the cells were transferred to a fresh culture medium and were allowed to grow for 1 h before the addition of 5-aminouracil to a final concentration of 0.5 mM. After 24 h, the cells were washed six times with fresh medium for 1 h, transferred to fresh medium without 5-aminouracil, and allowed to grow under aeration conditions. Samples of 0.5 mL were taken from these flasks at different times so as to observe cells under a microscope to check for synchronization of the cell cycle with different fluorescent staining procedures. Control cultures were prepared in the same manner but without 5-aminouracil.

### 2.6. Fluorescent Microscopy

A 100 μL aliquot of algal cells in a culture medium were stained for 30 min at room temperature with either 1 μL of a 1:100 stock solution of SYBR Green I Nucleic Acid Gel Stain in DMSO, (Invitrogen S7563, Invitrogen, Carlsbad, CA, USA) which stains DNA in green (it is maximally excited at 497 nm, but also has secondary excitation peaks at ~290 nm and ~380 nm; the fluorescence emission is centered at 520 nm), or with 1 μL of a 1 mg/mL DAPI (4’,6-diamidino-2-phenylindole; Sigma D9542, Sigma, St. Louis, MO, USA) solution, which stains DNA blue (DAPI alone, Ex = 340 nm, Em = 488 nm; DAPI–DNA complex, Ex = 364 nm, Em = 454 nm) and ortho-polyphosphate (poly-P) yellow-green [76,77]. The samples were then centrifuged at 1476× *g* for 10 min at room temperature and washed three times with 100 µL of deionized water. The cells were then observed with an Olympus BX50 fluorescence microscope (Olympus, Center Valley, PA, USA) at 60×/1.0, UPlan-FL (α−0.17) coupled to a Lumenera Infinity digital camera (Lumenera, Ottawa, ON, Canada), with a UV lamp for illumination and equipped with an U-MWIBA2 filter (excitation 460–490 nm, emission 510–550 nm). Chloroplasts were detected by chlorophyl autofluorescence at 680 nm [78,79]. Micrographs were taken using a Lumenera Infinity digital camera. Processing of the images was done with Image Pro premier 9.1 software (Media cybernetics, Rockville, MD, USA) and ImageJ [80] was used to overlay the images.

### 2.7. Multiphoton Microscopy

A 1 mM stock solution of MitoTracker Red FM (Ex/Em: 581/644 nm) (Invitrogen M22425, Invitrogen, Carlsbad, CA, USA) was prepared in DMSO [81], and 1 µL of this solution was used to stain 100 µL of algal cells for 30 min at room temperature in the dark. The staining process was completed by three cycles of centrifugation at 6242× *g* for 4 min and suspension in deionized water at 1:100 dilution. A 10 µL sample of the stained cells was taken for microscopy. The cells were set on a glass slide (Clear slides 75 × 25 mm, Catalog Number 2947-75×25) and slightly pressed with a high performance Zeiss cover slip (D = 0.17 mm ± 0.005 mm, refractive index = 1.5255 ± 0.0015, Abbe number = 56 ± 2; High Perf Coverslips (1000) 18 × 18 mm² 1.5H, Catalog Number: 444962-0000-000) to be observed with a multiphoton microscope system (LSM 880–NLO, Zeiss, Carl-Zeiss-Stiftung, Stuttgart, Germany) with an immersion objective of 60×/1.4, NA α−0.17, Zeiss Plan NEOFLUAR. The MitoTracker was excited at 543 nm and the emission was observed between 590–730 nm. All of the samples were captured in 15 micrographs (with three repetitions) at 1.4X–1.4Y zoom, equivalent to 9291 µm^2^ of the scanned area. All of the images were captured in CZI format at 1131 × 1131 pixels and RGB color.

## 3. Results

### 3.1. Identification and Bioinformatic Analysis of RBR and CDK Proteins

In order to begin understanding the aspects of cell cycle regulation in *Botryococcus braunii*, we used bioinformatics searches to identify the retinoblastoma (RBR) and cyclin-dependent kinase (CDK) sequences from the A and B races of this alga. These sequences were identified in the A race from the transcriptome sequence and from the B race using the genomic sequence [55]. This search resulted in identifying one RBR protein from each race (Appendix A). The alignment of these proteins with the full amino acid sequences of the RBR proteins from *C. reinhardtii*, *O. tauri*, *G. pectorale*, female and male variants of *V. carteri f nagariensis*, and *A. thaliana* was done, as well as alignment of the A and B domains of the RBR proteins (Appendix A). There was remarkable similarity among all sequences at the end of the RBR-A domain (amino acids 500–600), as well as around the LXCXE motif-binding site and to the end of the RBR-B domain (amino acids 750–900). A phylogenetic tree of the RBR proteins was created (Appendix A) showing that the RBR of both *B. braunii* races are more related to that of *O. tauri* than to the other algae.

Phosphorylation sites on RBR proteins are very important for their function. Non-phosphorylated RBR inhibits the cell cycle progression until the cell is ready to divide. At that point, RBR is inactivated through phosphorylation by CDKs at particular sites, allowing cells to enter into the cell cycle [68]. Potential phosphorylation sites on the RBR proteins used in this study were identified using the NetPhos phosphorylation site search engine (Appendix A). The consensus phosphorylation site for CDKs is S/T-P-X-K/R, where X is any residue, and the minimum sequence required is S/T-P [69]. It was reported that using the minimum sequence requirement for CDK of S/T-P, a total of 16 potential phosphorylation sites were found in three main areas within the human RBR, and the phosphorylation of all 16 consensus sites was confirmed through a combination of site-specific methods like reverse mutational analysis using site-directed mutagenesis and mass spectrometry [82]. So, potentially all phosphorylation sites are able to be active. Interestingly, of all the RBR proteins we analyzed RBR from *B. braunii*, race A had the most potential phosphorylation sites, with 23 sites containing the minimum consensus sequence and 6 specific CDK sites (Table 1). Race B RBR had 20 minimum sequence sites and 4 specific CDKs sites, *O. tauri* had 16 minimum sequence sties and 8 specific CDKs sites, *C. reinhardtii* had 17 minimum sequence sites and 4 specific CDKs sites, *G. pectorale* had 20 minimum sequence sites and 5 specific CDKs sites, *V. carteri f. nagariensis* (m) had 14 minimum sequence sites and 2 specific CDKs sites, *V. carteri f. nagariensis* (f) had 15 minimum sequence sites and 3 specific CDKs sites, and *A thaliana* had 16 minimum sequence sites and 9 specific CDKs sites (Table 1). The actual in vivo phosphorylation of any of these sites in the A and B race RBR proteins will have to be confirmed experimentally to establish functional significance for RBR phosphorylation.

The search for *B. braunii* A and B race CDK sequences identified nine CDKs in race A and eight in race B. The sequence of these proteins around the PSTAIRE motif or variations on this motif were aligned with those from *O. tauri*, *C. reinhardtii*, *G. pectorale*, *V. carteri f. nagariensis*, and *A. thaliana* (Appendix A). The CDKs found in race A belonged to 7 classes (A, B, D, E, G, H, and I), of which two sequences CDKE (CDKE1 and CDKE2) and CDKH (CDKH1 and CDKH2) were identified (Table 2). The B race contained eight CDKs: one of each A, B, C, D, E, G1, H1, and I classes, but none of the F, G2, and H2 classes (Table 2). The PSTAIRE motif of all CDKAs was identical, the rest of the CDKs have PSTAIRE motif variations, some were identical among the different microalgae, but others were slightly different (Table 2).

All CDK sequences described here were phylogenetically analyzed. The CDKAs, CDKBs, CDKCs, CDKDs, CDKEs, and CDKGs of race A, race B, and *A. thaliana* are closely related and are in the same clade (Figure 1). CDKF1 is only found in *A. thaliana*, but is more related to the CDKI1 of both races of *B. braunii*. The CDKGs are peculiar because the sequences from both races of *B. braunii*, the two of *O. tauri*, and the one of *A. thaliana* are similar, but a completely different clade is formed by the other algae CDKGs, which are closer to the CDKAs (Figure 1).

### 3.2. Synchronization of B. braunii Cultures

Next, we wanted to look for differences in the accumulation of the transcripts for RBR and the CDKs during the different phases of the cell cycle. A strong coordination of transcript expression with the cell cycle was shown using synchronized cultures [83].

Thus, we attempted to synchronize the cultures of the A and B race sof *B. braunii* with 5-aminouracil, as previously described [25]. In that study, synchronization was assessed by the formation of lipid bodies/vacuoles observed through light and fluorescence microscopy, and looking for the septum formation and the presence of swollen cells [25]. We did not use those assessment parameters as those observations were too general to distinguish among specific phases of the cell cycle. Other studies validated the synchrony of cells by evaluating the cell density and cell volume with a cell counter and size analyzer [83]. Such procedures are very complicated for *B. braunii*, because there are 50–100 cells per colony, and the colonies are surrounded by hydrocarbons and polysaccharides [2]. So, in order to assess cell synchronization, we used fluorescent staining of the nuclei with SYBR Green, as described for various species of *Chlorophyta*, *Rhodophyta*, and *Cyanophyta* [50,77,84]. Unfortunately, we were unable to synchronize *B. braunii* cells of either race. Although cells in other colonial algae like *Volvox* or *Haematococcus pluvialis* are able to be synchronized [84,85], there are no other reports of synchronization for *B. braunii*. A difference between our study and that of Hirose et al. [25] is the strain of the A race used; they used a race A strain from the Culture Collection of Algae at the University of Texas (UTEX 2441), while we used the A race Yamanaka strain [5].

### 3.3. Microscopy of Cells under Growing Conditions

While viewing *B. braunii* cells by microscopy in this work, we observed that under described growing conditions, some cells of the A race appeared to be multinuclear. This was not observed in race B. Upon further inspection, in samples taken between 15 to 20 days after inoculation, some race A colonies contained cells with a different morphology having an increased cell size and without chlorophyll in comparison with most of the pear-shape normal green cells, besides some red-orange cells perhaps containing carotenoids (Figure 2). After DAPI staining, which stains DNA blue and ortho-polyphosphate (poly-P) yellow green [76,77], and chlorophyll autofluorescence, bigger cells did not have chlorophyll (Figure 3A,B) and appeared to be multinucleated (Figure 3C,D). Poly-P were in the center of the colony and in some of the other cells.

To check the multinucleated cells, SYBR Green, which strongly binds double-stranded DNA, as well as single-stranded DNA and RNA but with lower affinities [86,87], and DAPI staining were done, and four or more nuclei were observed in cells without chlorophyll (Figure 4). However, the colonies and cells from race B cultures grown under the same growing conditions all looked similar (Figure 5A). All of the race B cells showed normal chlorophyll autofluorescence (Figure 5C,D), and all appeared to have a single nucleus after SYBR Green staining (Figure 5B,D). We do not know of any report describing this multinucleate situation in any race of *B. braunii*, but multinuclear phases have been reported in other microalgae. For example, multinucleate cells were observed during cell-cycle progression in *Scenedesmus *quadricauda** [50], and in *Raphidocelis subcapitata*, multinucleated stages have been seen under stress conditions [88].

### 3.4. Mitochondria Location

An intriguing trait that attracted our attention was the absence of chlorophyll in the multinucleated cells of the A race (Figure 3C,D and Figure 4C,D,G,H). Without chlorophyll, there is no photosynthesis; if these cells are not actively producing sugars from photosynthesis to be used in ATP production, where is the cellular energy coming from for metabolic requirements and to proceed with the cell cycle? The other organelles able to supply energy are the mitochondria. Besides ATP production from sugar, it has been proposed that enzymatically controlled hydrolysis of microparticles of Ca^2+^ and poly-P (Ca-poly-P-MP) via alkaline phosphatase (ALP) generates inorganic phosphate (P_i_), Ca^2+^, and metabolic energy that, in the presence of adenylate kinase (ADK), is stored in ADP and ATP [89]. To check the mitochondrial location and functionality, we used the MitoTracker Red FM, which is a fluorescent dye labeling mitochondrion within live cells [90]. This stain was used together with DAPI staining to confirm the nucleus location. After staining, the samples were analyzed using a multiphoton microscope. Race A multinucleate cells (Figure 6A) had a large mitochondrial active net that resembled the morphology of *B. braunii*, which is cup-shaped [15]. A different arrangement was seen in race B cells where the single nucleus was also near the mitochondrial net, but the mitochondrial net was smaller than in race A and did not follow the chloroplast outline (Figure 6B). These results confirm our previous observations of multinucleated cells in *B. braunii* race A (Figure 3C,D and Figure 4C,D,G,H), and cells with just one nucleus in *B. braunii* race B (Figure 5B,D), as well as the presence of an active mitochondrial net in both *B. braunii* races.

### 3.5. Other DAPI Stained Structures

It is recognized that DAPI stains the nuclei because it binds strongly to adenine–thymine rich DNA regions [76]. However, DAPI also stains other structures such as polyphosphates [77,91]. Inorganic polyphosphate (poly-P) is an ortho-phosphate (Pi) polymer that can be found in all living organisms [92], including *B. braunii* [93]. Vítová et al. [77] noted that DAPI stains *S. quadricauda* nuclei and nucleoids, and poly-P, which can be viewed at blue and yellow-green emission wavelengths, respectively. During DAPI staining of *B. braunii* cells grown for 15 days, we detected several nuclei, nucleoids, and poly-P, and differences in the form and distribution of these structures between the two races were observed (Figure 7). Most of the race A cells showed nuclei and nucleoids in blue and several poly-P in a yellow-green fluorescence (Figure 7A). In the race B cells grown under the same conditions, DAPI staining showed one nucleus per cell and less poly-P (Figure 7B).

### 3.6. Cell Division States in Cells of B. braunii Race A Colonies

As we saw previously, *B. braunii* race A cultures observed under growing conditions showed normal cells with intact chloroplasts, some bigger cells without chlorophyll, and some red-orange cells perhaps containing carotenoids (Figure 2). In a close-up of Figure 2 after staining with SYBR Green and DAPI, and analyzing the chloroplast autofluorescence (Figure 8), we observed cells with one nucleus showing fluorescent chloroplasts (Figure 8B, triangles 1, 2, and 3), some cells with less chlorophyll and blue nucleoids where the nucleus was less defined and smaller than the cells in triangles (Figure 8B, rectangles 4, 5, and 6), and bigger cells without chlorophyll with several nuclei stained with SYBR Green and some putative nucleoids (Figure 8, pentagons 7, 8, and 9). It has been reported that carotenoids may be detected by fluorescence with an excitation spectrum between 450–500 nm and an emission between 500–700 nm [94,95], and a mixture of chlorophyl and carotenoids looks yellow–orange (Figure 8B, circle and ellipse) [79]. Other pigments that may be detected by fluorescence are pheophytins (550–600 nm), a degradation product of chlorophyll when it loses the Mg^2+^ ion [96]. A possible explanation for this image is that the cells in race A colony are in different states of cell division. Some with chlorophyll and one nucleus may be under normal binary fission (triangles); others may be in transit to multiple fission where the nucleus is smaller, several nucleoids appeared, and chlorophyll is disappearing (rectangles); and other multinucleated bigger cells without chlorophyll may be under multiple fission (pentagons). Some of these bigger cells may be forming a coenobium-like structure, such as those at rectangle 6 including the cell in pentagon 9 in Figure 8B.

## 4. Discussion and Conclusions

As mentioned earlier, the cell doubling time of *B. braunii* is slow in comparison to other species of microalgae. The cell doubling time (*t_d_*) is the time required for cells to double in number. When the relative growth rate (*μ*) is constant, the population undergoes exponential growth and has a constant *t_d_*, which can be calculated directly from *μ* [97]. The *t_d_* of any organism depends on several factors, like the growth conditions, strain, nutrients, temperature, and light *μ* and hence *t_d_* are related to the cell cycle of the organisms [98]. In this work, we compared the sequences of the cell cycle regulators RBR and CDKs from *O. tauri*, an unicellular microalga that reproduces through binary fission; *C. reinhardtii*, an unicellular microalga that reproduces through multiple fission; *G. pectorale*, a colonial microalga that also reproduces through multiple fission; and *V. cartieri nagariensis* (male and female), a multicellular microalga that reproduces through multiple fission, along with the currently available data of the colonial *Botryococcus braunii* race A (Yamanaka strain) and *B. braunii* race B (Showa or Berkeley strain). Previous information about the cell cycle of *B. braunii* assumed a binary fission type of cell division [14,15,25].

The bioinformatic data in Table 1 show the possibility of 20 potential CDK phosphorylation sites in RBR of race B, and 23 in race A. Thus, *B. braunii* appears to be the microalga with the most putative phosphorylation sites among the studied organisms in this work. The race A and B CDK phosphorylation sites in RBR seem more similar to *G. pectorale* with 20 sites, and *V. carteri* (15 and 14 for f and m, respectively). However, the phylogenetic analysis (Appendix A) suggests that the RBR of both *B. braunii* races are more similar to that of *O. tauri*. It was reported that the phosphorylation of individual RB residues (SP or TP) caused major changes to the RB protein structure [99]. However, a single phosphorylation event has been shown to modify specific binding domains without compromising the overall integrity of the protein, leaving the other binding surfaces intact [100]. Experiments with synchronized human cells indicated that CDKD phosphorylated RB in the G1 phase, while Cyclin E/CDK2 hyper-phosphorylated RB at the G1/S transition, and this inactivated RB’s nuclear functions [99]. This analysis showed that RB isolated from asynchronously dividing cells was in two general states. A hyper-phosphorylated RB state generated by cell cycle progression seems to be phosphorylated on all 14 CDK phosphorylation sites, but hypo-phosphorylated RB, mainly mono-phosphorylated RB (mP-RB), can occur at many sites and perhaps at all 14 known CDK phosphorylation sites. This suggests that “active” RB may be the integrated effect of many mP-RB isoforms, perhaps as many as 14 different mP-RBs [99]. In this model of asynchronously dividing cells, multiple active mP-RB isoforms inhibit interactions with specific proteins, but promote interactions with others, suggesting that different aspects of RB function might be controlled [101]. The presence of many phosphorylation sites in the RBR of *B. braunii* race A (Table 1) may allow for more mP-RBR isoforms that are able to interact with many cellular targets during the early G1 phase.

The PSTAIRE motif is responsible for binding cyclins, and variations of this motif in the different CDKs has been used to classify the types of CDKs. In general, it is accepted that in plants, CDKA;1 controls the G1/S and G2/M transition, CDKBs help maintain the 2C ratio in the cell, CDKCs control cellular differentiation through RBR inactivation, CDKE;1 is involved in cell expansion in leaves and in the cell-fate specification in floral organs, CDKDs and CDKF;1 interact with CDKDs being activated through phosphorylation by CDKF;1 and they have roles in accelerating S-phase and the overall growth of cells, and CDKGs help in cell differentiation [44]. Neither CDKHs or CDKI have been identified in land plants, but they are present in algae with multiple fission such as *C. reinhardtii*, *G. pectoral*, and *V. carteri nagariensis*, and apparently in both *B. braunii* races A and B too (Table 2). The full role of all these CDKs in controlling the algal cell cycle is currently unknown. The phylogenetic analyses (Figure 1) show interesting similarities between CDKF;1, which is only present in plants, and CDKI1 present in multiple fission algae as mentioned previously, but the putative CDKIs from *B. braunii* are closer to the CDKF;1 of plants. In addition, CDKGs are in two separated clades, one including *C. reinhardtii*, *G. pectoral*, and *V. carteri nagariensis*, and the other including *O. tauri*, *A. thaliana*, and both proposed for *B. braunii* races A and B (Figure 1).

These results propose differences in the potential phosphorylation sites of RBR, in the CDK types, and hence in the respective variations of PSTAIRE motifs within the *B. braunii* race A and race B CDKs, which could end in differences between the cell cycles of these two races. It is known that different RBR phosphorylation states may control different and specific processes, but at the same time respond to the general culture conditions [101]. This could be the case for metabolically active *B. braunii* race A cultures, where most cells seem to be in different stages of the cell cycle process and the race A RBR forms could contain up to 23 different mP-RBR proteins, three more than in race B (Table 1). It has been reported that the phosphorylation state of pea RBR has a role in the regulation of the E2F-mediated gene expression involved in the G1/S transition and DNA replication during the dormancy-to-growth transition [102]. In addition, RBR may have a novel role in specifying the *Volvox carteri* gender [70]. Race A seems to have one additional CDKE2 compared to race B, as well as one putative CDKH2, not found in any other organism studied in this work (Table 2). These putative proposed enzymes could have a role in the cell cycle of these races. However, further experimental analyses must be conducted to determine the function of the putative proteins.

Unfortunately, we were unable to synchronize the *B. braunii* cultures to analyze the gene expression of the CDK and RBR proteins, but during the microscopic observations, we found other differences in the cell division processes of these *B. braunii* races. We observed some *B. braunii* race A multinucleated cells without chlorophyll (Figure 2, Figure 3 and Figure 4), while race B cells showed all cells with a single nucleus and chlorophyll after 15 days of inoculation under normal growing conditions (Figure 5). As mentioned previously, multinuclear phases are common in other microalgae that divide by multiple fission processes [49,50]. This mechanism is considered normal in the Chlorococcales and Volvocales orders, which are fast-growing algae [103]. Although *B. braunii* has some characteristics of Volvocales, like the extracellular matrix (ECM) and coenobium-like structure (Figure 8) [32,38], multiple fission has not been reported in *B. braunii*. Due to the production of useful hydrocarbons, most of the reports on the factors influencing the growing of *B. braunii* have been focused on strategies to increase lipid production [104,105,106,107,108]. Probably, all these procedures modify the natural cell cycle of the alga and up to now, all races have been assumed to have binary fission [14,15,25]. Multiple fission of Volvocales is favored under metabolically active growing conditions such as optimal light, nutrients, and temperature [49,109], and in our case, race A showed multinucleated cells and putative multiple fission when grown at optimal conditions 15 days after inoculation. In *B. braunii* race A, but not in race B, sufficient access to nutrients and/or CO_2_ may be a factor regulating this transition. Different from other algae, most recognized races of *B. braunii* produce large amounts of long chain hydrocarbons that require a significant demand of carbon flux [105,106]. The growing conditions may allow most of the cells to be in close contact with the dissolved CO_2_ as well as to have sufficient light for photosynthesis, increasing the metabolic rate. This may be a favorable condition for some cells on race A colonies for inducing division by multiple fission.

Multinuclear cells of race A showed an absence of chlorophyll (Figure 3, Figure 4 and Figure 8), and it has been reported that after autospores are formed during multiple fission in the unicellular green alga *Desmodesmus subspicatus*, they must reorganize their photosynthetic apparatus before achieving the maximum photosynthetic capacity [110]. Similarly, in *S. quadricauda* a decline in photosystem II activity was reported in preparation for the second CP, and cells showed a strong decline in photosynthetic activity, which was minimal when the protoplast division started [111]. This fits with our observation of the absence of chlorophyll in multinucleated cells of *B. braunii* race A; however, we were unable to track the division process of these multinucleated cells to confirm chlorophyll production along the division process.

If the required energy for metabolic activities and cell growth is not obtained from functional chloroplasts, it may come from an active mitochondrial net, which can be as big as the chloroplast as shown in race A (Figure 6A). Interestingly, in race B, the mitochondrial net is smaller than in race A (Figure 6B). In absence of chlorophyll, the energy may come from poly-P producing ATP and inducing a gene expression for cell differentiation [89,112]. It is known that poly-P is essential for growth, for responses to stresses, as well as for virulence of pathogens [113,114]. These polymers are stable at over a range of temperatures, pH, and oxidative states; are considered a source of energy; can act as a phosphorylating agent for alcohols, sugars, nucleosides, and proteins; can activate the precursors of fatty acids, phospholipids, and proteins; and supply material for nucleic acids during cell division [115,116]. Vítová et. al. [77] mentioned that DAPI may stain *S. quadricauda* nuclei and nucleoids in blue and poly-P in yellow–green. Interaction of poly-P with DAPI shifts the emission maxima from 456 nm for DNA to 526 nm for poly-P [117]. Thus, nuclear DNA and poly-P stained by DAPI can be distinguished by emission detection. Putative nucleoids could be seen in both *B. braunii* races, but race A seems to have more poly-P than race B (Figure 7A,B). However, if the blue DAPI stained spots were nucleoids attaching the dye to the minor grove of A-T [76], SYBR Green staining should make them greenish [86,87] if DNA is present, as in Figure 4. It is known that DAPI can also bind to RNA (emitting light at *c*. 500 nm) and to Pi-rich compounds, including inositol polyphosphates and poly-P (emitting light at *c*. 550 nm). Thus, DAPI cannot distinguish between inorganic and inositol polyphosphate molecules [118]. So, in our case, the blue DAPI staining shown in Figure 3, Figure 7 and Figure 8B may be due to other kinds of poly-P polymers, perhaps not so dense or large as those with the yellow-green color. Another possibility is that they are RNA, a kind of adenine and/or thymine storage or other kind of poly-P, with the emission maxima around 456 nm and having perhaps some role in a transition from binary to multiple fission in race A cells. 

There are two mechanisms of multiple fission in microalgae: the grouped one that does not present multinucleated forms, such as *C. reinhardtii* [119], and the consecutive one in which there are multinucleated cells, such as *Scenedesmus* sp. and *Chromochloris* sp. [77,120]. Multinucleated cells are present in both algae, and the nuclei in mononucleated cells were bigger than the nuclei of the multinucleated cells [120]. In this context, our results are similar and suggest that *B. braunii* race A colonies may have cells that undergo multiple fission in multinucleated cells. It is important to emphasize that all these observations were done with *B. braunii* race A, Yamanaka strain, and race B, Showa strain. We do not know if race B, Showa strain, would show multinuclear cells under different growth conditions. In addition, it is unknown if these are the only condition where the races used in this study show this behavior, and if other *B. braunii* strains and races behave similarly under other conditions. Our results allow us to design more specific experiments to elucidate the factors and mechanism related to cell division in *B. braunii*. In spite of the described differences in the RBR phosphorylation sites between race A and race B, as well as the different number and types of CDKs, there are no results suggesting these differences are related to the presence of multinuclear cells in race A, or with the type of cell division of these strains. The study of the cell cycle of other algae showing binary and multiple fission processes, like *R. subcapitata* or *Chromochloris zofingiensis* [120], should give more hints on the role of RBR and CDKs in these processes and their regulation [88].

## Figures and Tables

**Figure 1 biomolecules-11-01463-f001:**
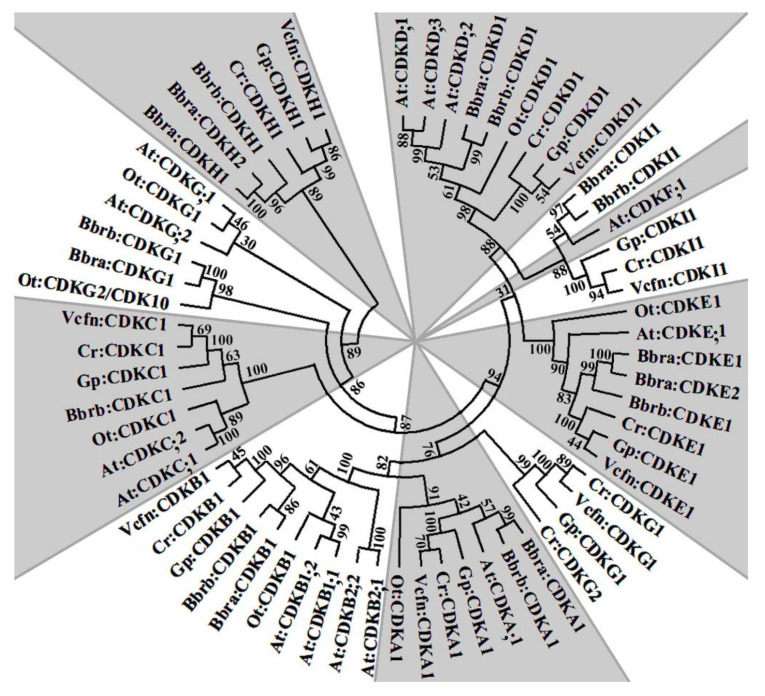
Phylogenetic tree of microalgae CDKs by maximum likelihood analysis. Bbra and Bbrb—*B. braunii* race A and B, respectively; Cr—*C. reinhardtii*; Ot—*O. tauri*; Gp—*G. pectorale*; Vcfn—*V. carteri* f. *nagariensis*; At—*A. thaliana* plant.

**Figure 2 biomolecules-11-01463-f002:**
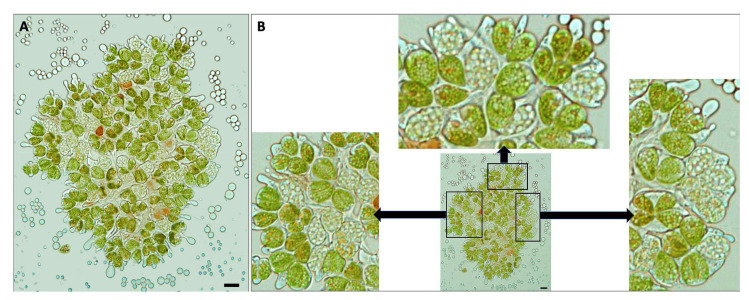
Bigger cells without chlorophyll in *B. braunii* race A colony. (**A**) Colony of *B. braunii* race A in bright field, surrounded by secreted liquid hydrocarbons containing several cells without chlorophyll bigger than the normal green cells. (**B**) Close-up of three areas from same colony in bright field, where some bigger cells without chlorophyll are clearly observed into the colony. Bar = 10 µm.

**Figure 3 biomolecules-11-01463-f003:**
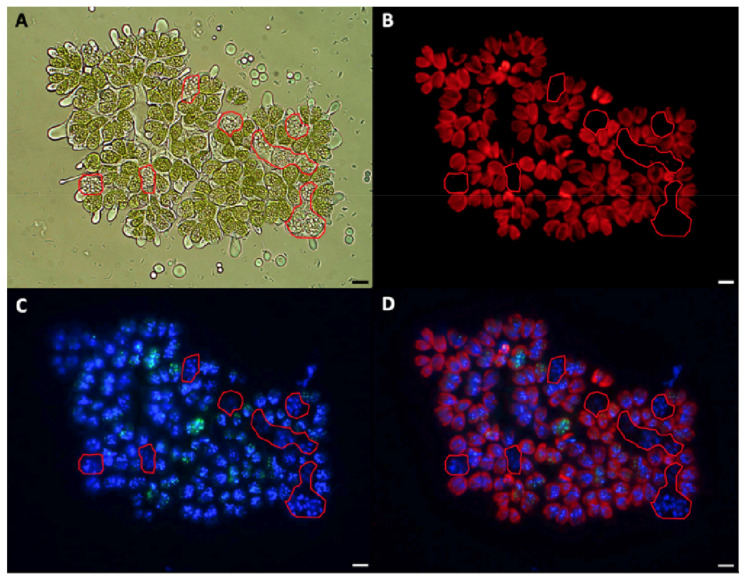
Fluorescent detection of cells without chlorophyll in *B. braunii* race A. (**A**) A colony of *B. braunii* race A in the bright field, surrounded by secreted liquid hydrocarbons containing several cells without chlorophyll bigger than the normal green cells, enclosed in a red line. (**B**) Same colony showing chloroplasts autofluorescence. (**C**) Same colony stained with DAPI. (**D**) Merging of (**B****C**). The same cells enclosed in the red line are in (**A**–**D**). Bar = 10 µm.

**Figure 4 biomolecules-11-01463-f004:**
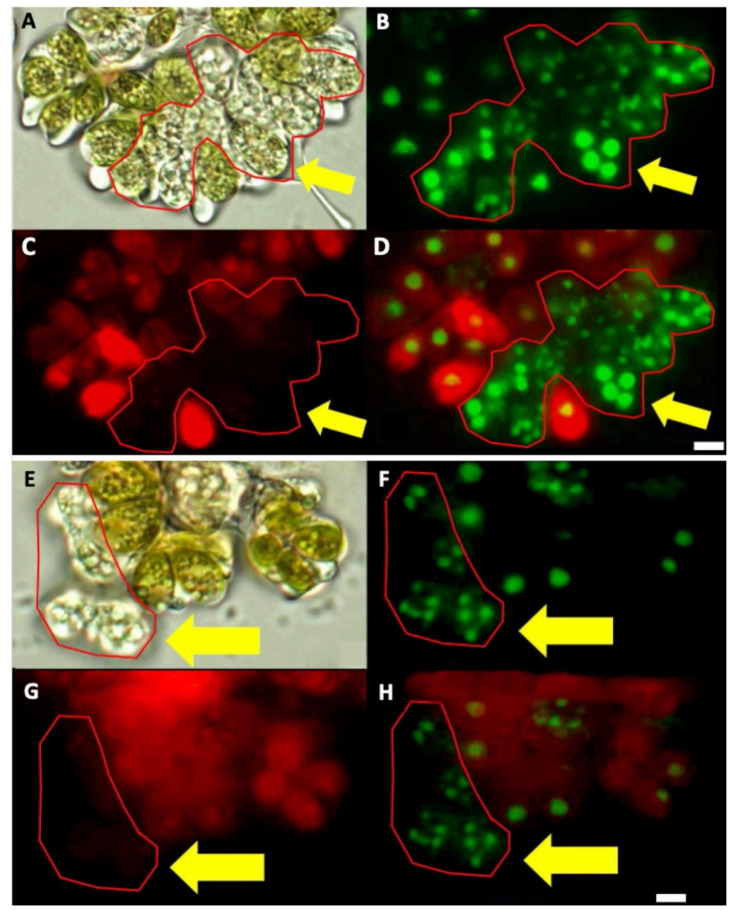
Multinucleated cells without chlorophyll in *B. braunii* race A. (**A**,**E**) Specific areas of two different colonies of *B. braunii* race A in bright field, containing several cells without chlorophyll, enclosed in a red line. (**B**,**F**) Same areas stained with SYBR Green–DAPI. (**C**,**G**) Same areas showing absence of chlorophyll autofluorescence. (**D**,**H**) Merging of (**B**,**C**) and (**F,G**). Yellow arrows indicate the multinucleated cells. Bar = 5 µm.

**Figure 5 biomolecules-11-01463-f005:**
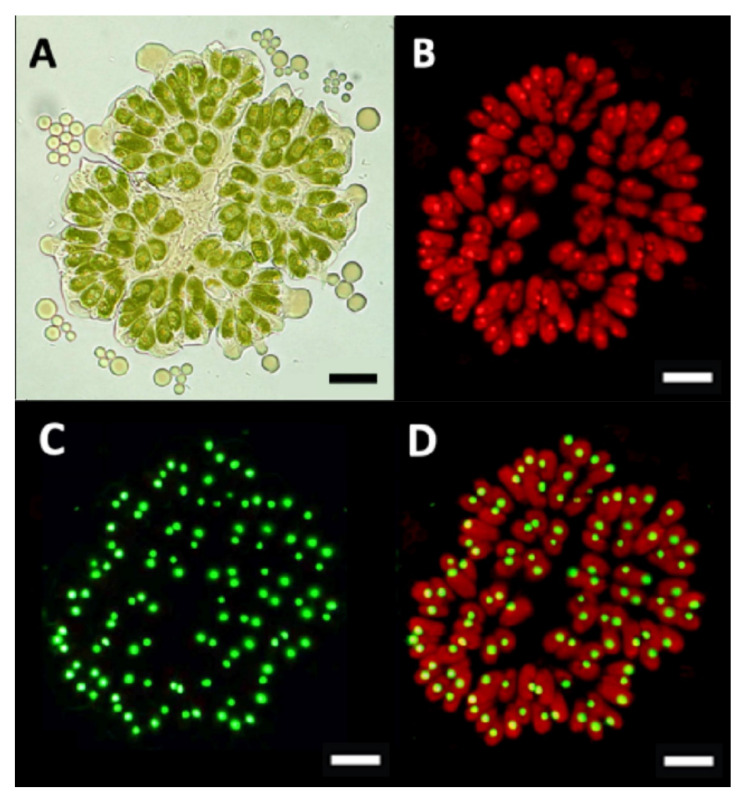
Colony of *B. braunii* race B. (**A**) Colony in bright field, all cells have chlorophyll. (**B**) All cells showing chloroplasts autofluorescence. (**C**) Same colony stained with SYBR Green, all cells are mononucleated. (**D**) Merging of (**B**,**C**). Bar = 10 µm.

**Figure 6 biomolecules-11-01463-f006:**
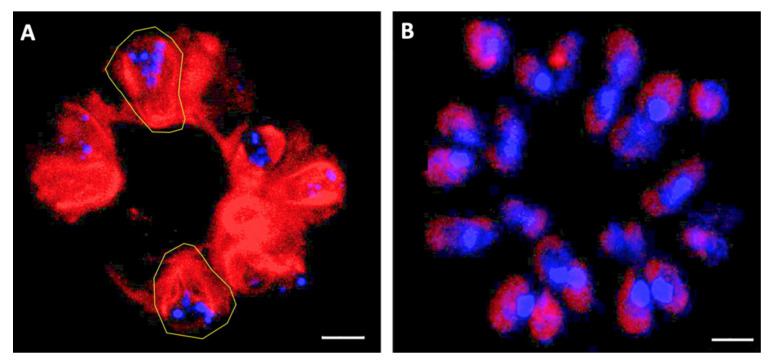
Mitochondrial net. Inner membrane electrical potential of active mitochondria in *B. braunii* races A and B were stained with MitoTracker and DAPI, and observed under a multiphoton microscope. (**A**) *B. braunii* races A colony, two mitochondrial nets similar to chloroplast forms containing several nuclei, are enclosed in a yellow line; (**B**) *B. braunii* races B colony, all cells are mononucleated showing the mitochondrial nets. Bar = 5 µm.

**Figure 7 biomolecules-11-01463-f007:**
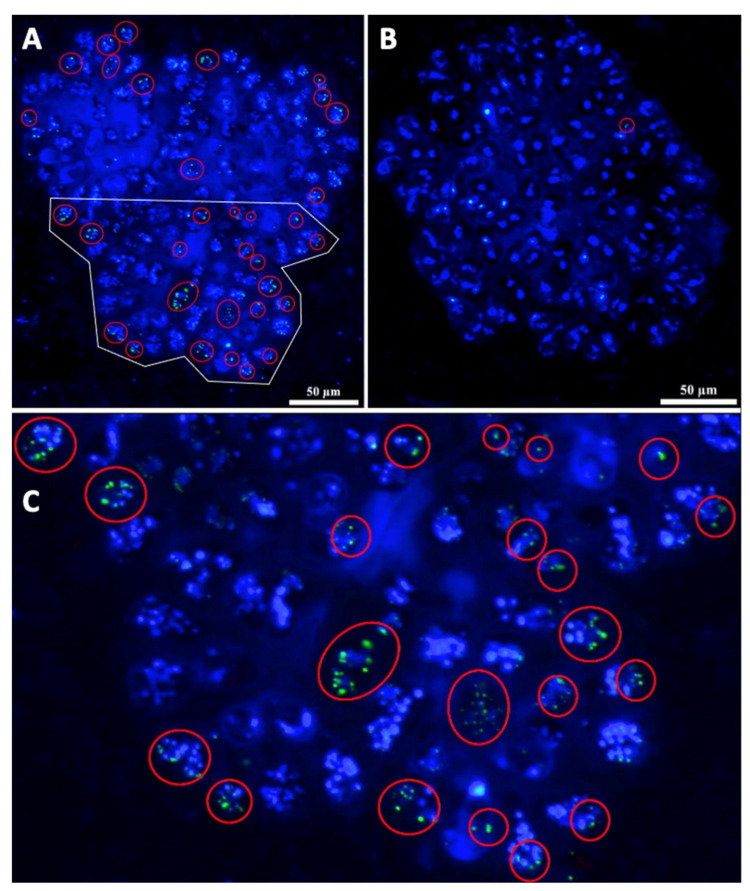
Nuclei, nucleoids, and poly-P stained with DAPI. (**A**) *B. braunii* race A cells show several nuclei and nucleoids in blue, and poly-P granules enclosed in red lines. (**B**) *B. braunii* race B cells show just one nucleus per cell, some with a few nucleoids in blue, and just two poly-P granules in one cell enclosed in a red circle. (**C**) Close-up of the area enclosed in a white line from panel B. Poly-P granules can be clearly seen enclosed in red lines.

**Figure 8 biomolecules-11-01463-f008:**
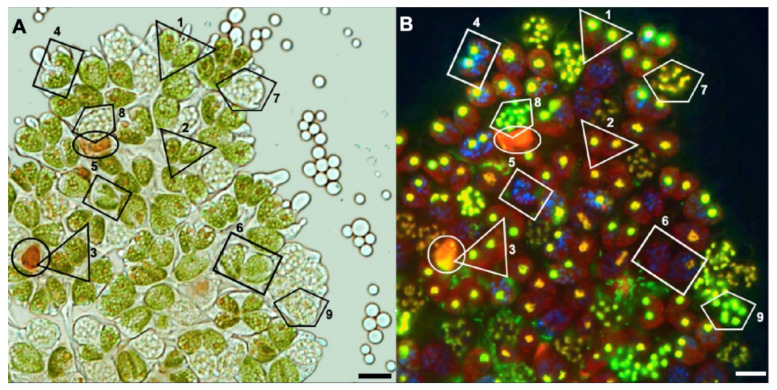
*B. braunii* race A cells under different cell division processes. Upper half of colony from Figure 2 showing (**A**) bright field of *B. braunii* race A colony showing pairs of cells in triangles 1, 2, and 3 and rectangles 4, 5, and 6 with chlorophyll, and pentagons showing bigger cells 7, 8, and 9 without chlorophyll. Circle and ellipse are cells with potential carotenoids as references. (**B**) Same colony area stained with SYBR Green and DAPI, and showing the chloroplast autofluorescence. Nuclei are yellow–green SYBR Green-stained, nucleoids are blue DAPI-stained, and chlorophyll is red fluorescent. Triangles 1, 2, and 3 show pairs of mononuclear cells with chlorophyll. Rectangle 4 shows a pair of cells with the nuclei surrounded by blue nucleoids and less chlorophyll than in the triangles. Rectangles 5 and 6 show small nuclei, nucleoids, and chlorophyll that almost disappeared. Pentagons 7, 8, and 9 show several nuclei in one cell without chlorophyll. Bar = 10 µm.

**Table 1 biomolecules-11-01463-t001:** Putative MAT3/RBR/RBR1 phosphorylation sites for cyclin-dependent kinases (CDKs).

Species	SP	TP	Total	Specific
*Ostreococcus tauri*	11	5	16	8
*Chlamydomonas reinhardtii*	11	6	17	4
*Gonium pectorale*	12	8	20	5
*Volvox carteri f. nagariensis* (f)	12	3	15	3
*Volvox carteri f. nagariensis* (m)	11	3	14	2
*Botryococcus braunii* race B (Showa)	9	11	20	4
*Botryococcus braunii* race A (Yamanaka)	14	9	23	6
*Arabidopsis thaliana*	14	2	16	9

SP—number of SP phosphorylation sites; TP—number of TP phosphorylation sites; Total—total phosphorylation sites; Specific—specific phosphorylation sites; f—female; m—male. The minimum sequence required for a phosphorylation site is S/T-P. A specific consensus phosphorylation site for CDKs is S/T-P-X-K/R where: S—serine; T—threonine; P—proline; K—lysine; R—arginine; X—any residue; from [69].

**Table 2 biomolecules-11-01463-t002:** Cyclin-binding PSTAIRE motifs of CDKs.

CDKs	*O. tauri*	*C. reinhardtii*	*G. pectorale*	*V. carteri f. nagariensis*	*B. braunii* Race B (Showa)	*B. braunii* Race A (Yamanaka)	*A. thaliana*
CDKA	PSTAIRE	PSTAIRE	PSTAIRE	PSTAIRE	PSTAIRE	PSTAIRE	PSTAIRE
CDKB	PSTALRE	PSTTLRE	PSTTLRE	PSTTLRE	PSTALRE	PSTALRE	PPTALRE PPTALRE PSTTLRE PPTTLRE
CDKC	PITAIRE	PITAIRE	PITAIRE	PITAIRE	PITAVRE	Nf	PITAIRE PITAIRE
CDKD	NFTAIRE	DPTALRE	DPTALRE	DPTALRE	NMTALRE	NVTALRE	NVTALRE NFTALRE NITALRE
CDKE	PAPTVRE	SPTAIRE	SPTAIRE	SPTAIRE	SPTAIRE	SPTAIRE SPTAIRE	SPTAIRE
CDKF	Nf	Nf	Nf	Nf	Nf	Nf	QSAFRE
CDKG1	PLTALRE	SDSTIRE	SDSTIRE	SDSTIRE	PVTSVRE	PVTAMRE	PLTSLRE
CDKG2	PLTSLRE (CDK10)	AASTLRE	Nf	Nf	Nf	Nf	PLTSLRE
CDKH1	Nf	PVTSIRE	PVTSIRE	PVTSIRE	PLTSIRE	PQTAIRE	Nf
CDKH2	Nf	Nf	Nf	Nf	Nf	PLTAIRE	Nf
CDKI	Nf	PDVVVRE	PDVVIRE	PDVVVRE	LSTRYPE	ENKPLKE	Nf

Nf—not found. CDKs nomenclature for microalgae: CDKA1, CDKB1, CDKC1, CDKD1, CDKE1, CDKE2, CDKG1, CDKG2, CDKH1, CDKH2, and CDKI1. CDKs nomenclature for *A. thaliana*: CDKA;1, CDKB1;1, CDKB1;2, CDKB2;1, CDKB2;2, CDKC;1, CDKC;2, CDKD;1, CDKD;2, CDKD;3, CDKE;1, CDKF;1, CDKG;1, and CDKG;2.

## Data Availability

The data presented in this study are available in the manuscript or Appendix A.

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
