# Peer review of "In Silico and Cellular Differences Related to the Cell Division Process between the A and B Races of the Colonial Microalga Botryococcus braunii"

_biomolecules, 2021, doi:10.3390/biom11101463_

Round 1
Reviewer 1 Report
none
Author Response
None comment from Reviewer 1. We thank for the revision the new version
Reviewer 2 Report
The revised manuscript by Morales-de la Cruz et al is greatly improved.
Most of my concerns/comments/suggestions have been appropriately addressed and dealt with.
I only suggest a careful editing for typos and style, especially in the newly added text. For instance, in the Introduction, line 74: “from 21 um to 70 mm”; should be 21 mm? Material and Methods, line 175-176: 680 or 280 nm? Line 401: “all appears” should be “all appear”. Line 519: “Bioinformatic data … shows” should be “show”. Line 522: “phosphorylation sites in RBR seems” should be “seem”.
Author Response
We appreciate the opinion of the reviewer about the improvement of the manuscript, the comments and suggestions to the previous version were wise and helped us to get a better manuscript.
Regarding the editing for typos and style in the newly added text:
Introduction, line 74: “from 21 um to 70 mm”; should be 21 mm?
The correction in the Introduction, line 74 is done: from 21 μm to 70 μm
Material and Methods, line 175-176: 680 or 280 nm?
The correction was done in line 175: …assessed when OD 680 nm reached…; and line 176: The OD 680 nm at.
Line 401: “all appears” should be “all appear”.
The correction is done: all appear
Line 519: “Bioinformatic data … shows” should be “show”.
The correction is done: Bioinformatic data in Table 1 show
Line 522: “phosphorylation sites in RBR seems” should be “seem”.
The correction is done: phosphorylation sites in RBR seem
This manuscript is a resubmission of an earlier submission. The following is a list of the peer review reports and author responses from that submission.
Round 1
Reviewer 1 Report
The authors present data about RBR and CDK´s molecules obtained in silico for two races (A & B) of the colonial microalga Botryococcus braunii. The rationale is the rather scarce knowledge of the cell cycle control in B. braunii, and that phosphorilation of RBR by CDK´s will trigger cell division as revealed in other alga (RBR pathway). The knowledge would be of max. interest in this hydrocarbon producing alga with a rather low doubling time as compared to other cultivated and (biotechnological) interesting alga.
The results focus on blasting of RBR´s A and B seqs. against RBR from other alga as well as the quantification of potential phosphorylation sites in their seqs, but no experiments support actual involvement of the RBR´s molecules and pathway in the control of the cell cycle in B. braunii, neither the expression of the putative in RBR and CDK´s genes could be even followed, as the required synchronous cultures could not be established. Complementarily, the authors presents changes in the cell structure during growth in vitro under different conditions that are very difficult to relate to the actual aim of the work.
I conclusion, and as the authors said “our results allow us to design more specific experiments to elucidate the factors and mechanism related to cell division in B raunii”, therefore the paper remains mainly speculative.
Therefore I regret to reject this paper in its present form.
Author Response
Reviewer 1
(Comment 1.1) The authors present data about RBR and CDK´s molecules obtained in silico for two races (A & B) of the colonial microalga Botryococcus braunii. The rationale is the rather scarce knowledge of the cell cycle control in B. braunii, and that phosphorilation of RBR by CDK´s will trigger cell division as revealed in other alga (RBR pathway). The knowledge would be of max. interest in this hydrocarbon producing alga with a rather low doubling time as compared to other cultivated and (biotechnological) interesting alga.
The results focus on blasting of RBR´s A and B seqs. against RBR from other alga as well as the quantification of potential phosphorylation sites in their seqs, but no experiments support actual involvement of the RBR´s molecules and pathway in the control of the cell cycle in B. braunii, neither the expression of the putative in RBR and CDK´s genes could be even followed, as the required synchronous cultures could not be established. Complementarily, the authors presents changes in the cell structure during growth in vitro under different conditions that are very difficult to relate to the actual aim of the work.
I conclusion, and as the authors said “our results allow us to design more specific experiments to elucidate the factors and mechanism related to cell division in B raunii”, therefore the paper remains mainly speculative.
Therefore I regret to reject this paper in its present form.
(Answer 1.1) We thank the comment of the reviewer, and we agree. We want to mention that we have been and are still looking for some procedure to synchronize the cultures to do some experiments to confirm the existence of reported CDKs, as well as the phosphorylation status of the RBR protein. Unfortunately, up to now we have not been successful. However, the bioinformatic data are valuable for other researchers interested in these proteins and are valuable to complete the genomic and transcriptomic data of these two races. Also, our results on microscopy are original and new, mainly because the cell cycle of these races of this alga has not been compared and studied previously. The presence of multinucleated cells in race A opens the possibility of binary and multiple fission strategies for this alga, which may help it adapt to different growth conditions. Race B does not show this trait, so there is a specific difference between these races besides the type of hydrocarbons produced. We agree that this difference must be experimentally demonstrated, and the synchronization of the cells in a culture is necessary. However, these observations are specific, and they are reported for the first time in our work.
Reviewer 2 Report
The manuscript by Morales-de la Cruz provides a comparative analysis of two strains of Botryococcus braunii with respect to (i) two sets of genes coding for proteins involved in the regulation of the cell cycle and (ii) some cytological features thought to be related to cell division.
Specifically, the authors identified potential homologs of RBR and CDKs in the two strains and discussed them in the context of other algal and plant counterparts. There are some unexpected findings regarding their phylogenetic relationships – but they might require additional analyses (see below). The authors also suggest that multiple fission can occur (under one specific set of conditions) in one strain, but not the other. However, it is not known whether these putative multinucleated cells divide; thus, in this reviewer’s opinion these observations might not be – at this time, enough to argue for a new type of cell division (and cell cycle regulation) in this strain.
Main suggestions/concerns:
- In the Introduction, it would be useful to include some information on the taxonomic/phylogenetic position of the species, to allow a better appreciation of the gene phylogenetic inferences presented. Also, additional information on the biology of the species itself would be useful – e.g., how many cells in the colony (is it a set number of cells?); are all cells reproducing (does each cell produces a new colony?; or does the colony grow and fragment?; what are autospores?; are they released and each makes a new colony?); any differences in cell number or colony organization between the two strains?; are cells flagellated?; is the species freshwater or marine?; is it strictly autotrophic? Such information would facilitate the understanding of the results and comparisons with Ostreococcus and Chlamydomonas….
- Doubling time: it is not clear if it refers to cell doubling or population doubling (ie, doubling of the number of individual colonies)…. In the introduction it is referred to as “cell doubling time” (and in Abstract as “ cell doubling rate”) but in the discussion it is stated that “The doubling time (td) is the period of time required for a population to double. “. It is not clear what the 1.3-6 days doubling time for B. braunii refers to…
- It is mentioned that “once cultures reached stationary phase (24-30 days after inoculation)”, aeration was stopped and cultures were kept for another 3 weeks. It would be useful to know what was the initial cell/colony density (and if it was the same for both strains) and what was the density in the stationary phase to assess if the cultures differed in cell density at the time of sampling, as such differences can influence may aspects of cell physiology. Also, not clear why/how the stationary phase was assessed – why 24-30 day after inoculation (why a range?)? If the two strains were not started at the same density or do not have the same growth rate, it is likely they reached stationary stage at different times.
- Why was the aeration stopped and why were the cultures kept for another 3 weeks? What was the goal/question? If the question was about comparing growth between aerated and non-aerated cultures, why not growing them in exponential phase under the 2 sets of conditions. In the stationary phase there are a number of other factors that affect cell biology and cell division…. So the observations reported are not directly due to (only) the lack of aeration…
- “For CDK sequences, this process identified 28 annotated CDK-encoding sequences in B race and after screening these sequences for the PSTAIRE motif, we selected eight CDKs to use for analysis. “; were these eight selected because they were the only ones that had the motif?
- Phylogenetic analyses: it would be useful to also perform maximum likelihood analyses, especially since some of the inferred relationships are unexpected, and some of the bootstrap values are low….. neighbour joining analyses are not ideal. Also, in Figure 1 – some information about the analysis would be useful (number of sites used, is the tree rooted?)
- “in samples taken between 15 to 20 days after inoculation under aeration conditions, some race A colonies contained cells with a different morphology having an increased cell size in comparison to most of the pear-shape normal cells (Figure 2A) “ – it would be good to indicate those cells in the colony image (panel A) as it is not clear which cells are larger… generally, it is hard to differentiate individual cells in panel A (cells seem almost fused/overlapping; is it a multilayered colony?). What are the blue spots in panel B? Also, in Figure 2 in addition to stating that the samples were taken from cultures after 15 days of growing it would be good to know the population density as that would likely influence cell physiology and cell cycle progression; and if the two strains were at different cell densities (and different points in the population growth curve) they might show different responses just due to such differences (not due to strain-specific differences)
- “To check if these differences were maintained along the culture cycle, the aeration of the B. braunii A and B race cultures was stopped when the cultures reached the stationary phase after three weeks of growth. Then, the flasks were kept in this non-aerated condition with the same photoperiod for three more weeks. “. Again – if the question was to investigate changes along the culture cycle, why stopping the aeration? Why including a new factor? Is this part of the culture cycle for hydrocarbon production???
- “After a further 21 days in the non-aerated condition, samples were carefully taken from the colonies on the surface. “. Why only from the surface? These colonies have likely experienced less the “non-aerated conditions” since they are at the medium-air interface….
- Figure 3: the nuclei in panel A seem a lot smaller than the nuclei in panel B… Is it possible that those are in facts nuclear fragments? Maybe as part of the activation of a stress-induced programmed cell death process? That would also explain the lack of chlorophyll/fluorescence… Mitochondria tend to change their morphology during stress/PCD too… Stress is also consistent with the presence of those red-orange cells in Figure 4D that the authors suggest are the result of stress…
- In Figure 4 panel A - are all cells multinucleate? And in panel D, again, the nuclei in “multinucleated cells” appear smaller than the nuclei in mononucleated cells … Where the light-green cells assessed for viability? They look dead or autophagic... Any suggestions as to why only some cells in the colony show this phenotype?
- “Many green algae such as carteri and C. reinhardtii are known to use palintomy, or multiple fission to proliferate [109]. In this context, our results suggest that B. braunii race A may have cells that undergo multiple fission in specific culture conditions. “. The two algae mentioned above do undergo multiple fission, but they are never multinucleated; cytokinesis immediately follows every mitotic event. So, they are not relevant to what the authors try to suggest here….
- “If multiple fission occurs in race A grown under the aerated and metabolically active condition, this situation may be similar to that in some Volvocales that were grown under illuminated periods that allows for metabolically active conditions [99, 42]. “. Not sure what is meant here, as Volvocales always divide via multiple fission…
Minor edits:
- In several places, including Figure 2, SYBR is misspelled (“SYBER”)
Author Response
Reviewer 2
(Comments)
The manuscript by Morales-de la Cruz provides a comparative analysis of two strains of Botryococcus braunii with respect to (i) two sets of genes coding for proteins involved in the regulation of the cell cycle and (ii) some cytological features thought to be related to cell division.
Specifically, the authors identified potential homologs of RBR and CDKs in the two strains and discussed them in the context of other algal and plant counterparts. There are some unexpected findings regarding their phylogenetic relationships – but they might require additional analyses (see below). The authors also suggest that multiple fission can occur (under one specific set of conditions) in one strain, but not the other. However, it is not known whether these putative multinucleated cells divide; thus, in this reviewer’s opinion these observations might not be – at this time, enough to argue for a new type of cell division (and cell cycle regulation) in this strain.
Main suggestions/concerns:
(Comment 2.1) In the Introduction, it would be useful to include some information on the taxonomic/phylogenetic position of the species, to allow a better appreciation of the gene phylogenetic inferences presented. Also, additional information on the biology of the species itself would be useful – e.g., how many cells in the colony (is it a set number of cells?); are all cells reproducing (does each cell produces a new colony?; or does the colony grow and fragment?; what are autospores?; are they released and each makes a new colony?); any differences in cell number or colony organization between the two strains?; are cells flagellated?; is the species freshwater or marine?; is it strictly autotrophic? Such information would facilitate the understanding of the results and comparisons with Ostreococcus and Chlamydomonas….
(Answer 2.1) We agree with the reviewer and appreciate the valuable observations which will improve the manuscript.
In response to the first observation, we have added the following text in the Introduction.
“According to the ITIS Report (Integrated Taxonomic Information System), Botryococcus braunii has a Taxonomic Serial No 6308, and the taxonomic hierarchy is: Kingdom Plantae, Subkingdom Viridiplantae, Infrakingdom and Division Chlorophyta, Class Trebouxiophyceae, Order Trebouxiales, and Family Botryococcaceae. A phylogenetic analysis of two isolates of B. braunii race A, one from race B, and one from race L, using the small nuclear sequence of the 18S rRNA, showed that the four B. braunii isolates form a monophyletic group that approximates the genus Choriscystis of the class Trebouxiophyceae. Weiss et al. (2010) generated a phylogeny of a B. braunii race B using the 18S rRNA sequence with the maximum likelihood (ML) method, which confirmed that it belongs to the Trebouxiophyceae class. Then, Kawachi et al. (2012) carried out a 18S rRNA phylogenetic study to relate the hydrocarbons produced by B. braunii, with the molecular phylogeny of the microalga. A phylogenetic tree using 31 axenic strains of the microalgae isolated from various regions of Japan was generated and gas chromatography-mass spectrometry was included. The results showed that all analyzed strains belonged to the Trebouxiophyceae class. This analysis revealed the existence of three major clades (I, II and III) within this class. Clade I consisted mainly of race A strains, clade II of race B strains, and clade III of a mixture of races L and S strains. Clades II and III were divided into two subclades (II1, II2, III1 and III2).
B. braunii race B (Showa) is a colonial green microalga present in brackish and fresh water. It has been found all around the world, produces long-chain hydrocarbons which reduce the density of large masses allowing them to float on the water surface. Each colony is composed of 50 to 100 pear-shaped non- flagellated cells held together in a hydrocarbon extracellular matrix (Weiss et al., 2012). Diversity of 10 strains of B. braunii race A isolated in different aquatic environments was studied regarding their morphology, hydrocarbon, and fatty acid profiles (Hegedűs et al., 2016). Irregular green, lax spherical small clusters of pyriform cells (14-15 mm), forming colonies of 21 mm to 70 mm size were observed. The size and number of lipid bodies were related to the cell cycle stage. Both races A and B are autotrophic and mixotrophic (Zhang et al., 2011). Noguchi et. al., 1999 describes the presence of autospores as two daughter cells into a mother cell wall, involved in the cell division mechanism in a colony of non-specified race of B. braunii. Weiss et. al., (2012) proposed that B. braunii colonies grow to a specific size, are then fragmented to produce smaller colonies which grow again. Suzuki et al., (2013) reports that during the growth stage of B. braunii race B (Showa), each cell divides into two daughter cells, but no information is given about the growth of the colonies. Single cells of B. braunii race B (Showa), released with glycerol, were unable to survive unless they were in a high concentration (>2x107 cells / mL) (Hou et. al., 2014).
-------
(Comment 2.2). Doubling time: it is not clear if it refers to cell doubling or population doubling (i.e., doubling of the number of individual colonies) …. In the introduction it is referred to as “cell doubling time” (and in Abstract as “cell doubling rate”) but in the discussion it is stated that “The doubling time (td) is the period of time required for a population to double. “. It is not clear what the 1.3-6 days doubling time for B. braunii refers to…
(Answer 2.2) In response to this observation, we refined the definition of cell doubling time in the Discussion: “The cell doubling time (td) is the time required for cells to double in number”. We think these changes clarify the definition.
-------
(Comment 2.3) It is mentioned that “once cultures reached stationary phase (24-30 days after inoculation)”, aeration was stopped, and cultures were kept for another 3 weeks. It would be useful to know what the initial cell/colony density was (and if it was the same for both strains) and what was the density in the stationary phase to assess if the cultures differed in cell density at the time of sampling, as such differences can influence may aspects of cell physiology. Also, not clear why/how the stationary phase was assessed – why 24-30 day after inoculation (why a range?)? If the two strains were not started at the same density or do not have the same growth rate, it is likely they reached stationary stage at different times.
(Answer 2.3) We appreciate this observation and in response we include in the Materials and Methods section, at the end of 2.1 Strains and culture conditions, the following sentence: “Growing of both races was followed spectrophotometrically starting with an initial inoculum of 0.03 average at OD 680 nm. The stationary phase was assessed when OD 280 nm reached a maximum and constant value along the time as mentioned before for each race. The OD 280 nm at stationary phase was 0.216 ± 0.028 at 24 days for race A, and 0.270 ± 0.05 at 30 days for race B. Race A grew little faster than race B.”
-------
(Comment 2.4) Why was the aeration stopped and why were the cultures kept for another 3 weeks? What was the goal/question? If the question was about comparing growth between aerated and non-aerated cultures, why not growing them in exponential phase under the 2 sets of conditions. In the stationary phase there are a number of other factors that affect cell biology and cell division…. So, the observations reported are not directly due to (only) the lack of aeration…
(Answer 2.4) Thanks for the wise observation and we can say that aeration was stopped to reduce the metabolic activity with the intention to synchronize the cultures, which has been the main problem we have for studying the cell cycle of this alga. To check synchronization, we took samples at 21 days after aeration was stopped and we did not find multinucleated cells, but cells were not yet synchronized. Unfortunately, we are still unable to get this synchronization, and we agree with the reviewer that many other factors affect cell biology in the stationary phase. Up to now, we do not have sufficient experimental evidence to demonstrate that lack of aeration is the reason for the disappearance of the multinucleated cells. So, we decided to eliminate Figures 2I, 2J, 2K, 2L, 2M, 2N, 2O and 2P. all of which are related to the non-aerated condition. Also, lines 25 to 29 of the Abstract were modified as follows: “Microscopic observations were done using several staining procedures. Under growing conditions, race A colonies, but not race B, showed some multinucleated cells without chlorophyll. An active mitochondrial net was detected in those multinucleated cells, as well as defined polyphosphate bodies. These observations suggest differences in the cell division processes between the A and B races of B. braunii”.
Also, at the end of section 2.1 Strains and culture condition section, the sentence (lines 136 to 139) was eliminated: “Once cultures reached stationary phase (24-30 days after inoculation), aeration was stopped, and flasks were kept in a non-aerated condition for three more weeks, at which time most of the cells were floating on the surface of the culture media”; and the last sentence (lines 139 to 140) was modified as follows: “Samples were taken at different times for microscopic observations”. Consequently, the whole paragraph in section 3.4. Microscopy of cells under non-aerated conditions was eliminated. The paragraph (lines 508 to 510) was modified as follows: “We observed some B. braunii race A multinucleated cells without chlorophyll, while race B cells showed all cells with a single nucleus and chlorophyll after 15 days of inoculation under normal growing conditions (Figure 2)”. Paragraph (lines 520 to 531) was modified as follows: “Multiple fission of Volvocales is favored under metabolically active growing conditions such as optimal light, nutrients, and temperature (Herron 2016; Ivanov et. al., 2016), and in our case race A show multinucleated cells and putative multiple fission when growing at optimal conditions 15 days after inoculation. In B. braunii race A but not in race B, sufficient access to nutrients and/or CO2 may be a factor regulating this transition. Different from other algae, most recognized races of B. braunii produce large amounts of long chain hydrocarbons which require a significant demand of carbon flux (Sakamoto et. al., 2012; Song et. al., 2012). The growing conditions may allow most of the cells to be in close contact with the dissolved CO2 as well as to have sufficient light for photosynthesis, increasing the metabolic rate. This may be a favorable condition to some cells on race A colonies for inducing division by multiple fission.” Finally, the paragraph on lines 567 to 575 was modified as follows: “Many green algae such as Scenedesmus sp. and Chromochloris sp. are known to use palintomy, or multiple fission, to proliferate (Umen & Olson, 2012). In this context, our results suggest that B. braunii race A may have cells that undergo multiple fission in the assayed culture conditions. It is important to emphasize that all these observations were done with B. braunii race A, Yamanaka strain, and race B, Showa strain. We do not know if race B, Showa strain, would show multinuclear cells under different growth conditions. Also, it is unknown if these are the only conditions where the races used in this study show this behavior, and if other B. braunii strains and races behave similarly under other conditions”.
-------
(Comment 2.5) “For CDK sequences, this process identified 28 annotated CDK-encoding sequences in B race and after screening these sequences for the PSTAIRE motif, we selected eight CDKs to use for analysis. “; were these eight selected because they were the only ones that had the motif?
(Answer 2.5) Correct, they were selected because they had the cyclin binding motif and the blastp results showed a higher percentage of identity with cyclin-dependent kinases (CDKs).
-------
(Comment 2.6) Phylogenetic analyses: it would be useful to also perform maximum likelihood analyses, especially since some of the inferred relationships are unexpected, and some of the bootstrap values are low…. neighbour joining analyses are not ideal. Also, in Figure 1 – some information about the analysis would be useful (number of sites used, is the tree rooted?)
(Answer 2.6) We welcome the reviewer's suggestion to improve this figure. The maximum likelihood analysis was performed, and we will replace Figure 1. In this way it better groups the CDKs of both races of B. braunii with that from O. tauri and A. thaliana. This fact was already shown in the original Figure 1 and the new figure, shows again that the amino acid sequences belonging to the CDKGs diverge in three clades. One clade is the Volvocales CDKG, another clade is the B. braunii CDKGs more related to the CDKG2 of O. tauri, and the last clade groups the CDKG1 of O. tauri, which is most related to the A. thaliana CDKG1 and CDKG2.
Also, the last sentence of the 2.4. Phylogenetic analysis section, (lines 201 and 202), was modified as follows: “The evolutionary history was inferred by using the Maximum Likelihood method and Poisson correction model (Zuckerkandl and Pauling, 1965). The bootstrap consensus tree inferred from 1000 replicates, was taken to represent the evolutionary history of the analyzed taxa (Felsenstein, 1985). Branches corresponding to partitions reproduced in less than 50% bootstrap replicates were collapsed. Initial tree(s) for the heuristic search were obtained automatically by applying Neighbor-Join and BioNJ algorithms to a matrix of pairwise distances estimated using the Poisson model, and then selecting the topology with superior log likelihood value. This analysis involved 63 amino acid sequences. All positions containing gaps and missing data were eliminated (complete deletion option). There was a total of 195 positions in the final dataset. Evolutionary analyses were conducted in MEGA X (Kumar et. al., 2018)”. Consequently, (lines 305 to 314) were modified as follows: “All CDK sequences described here were phylogenetically analyzed. The CDKAs, CDKBs, CDKCs, CDKDs, CDKEs, and CDKGs of race A, race B, and A. thaliana are closely related and in the same clade (Figure 1). The CDKF1 is only found in A. thaliana but is more related to CDKI1 of both races of B. braunii (Figure 1). The CDKGs are peculiar because sequences from both races of B. braunii, the two of O. tauri and that one of A. thaliana are similar, but a completely different clade is formed by the other algae CDKGs which are closer to the CDKAs (Figure 1).”. Lines 494 to 499 were modified as follows: “These results show differences in the potential phosphorylation sites of RBR, in the CDK types, and in the respective variations of PSTAIRE motifs within the B. braunii race A and race B CDKs, which could end in differences between the cell cycles of these two races. It is known that different RBR phosphorylation states may control different and specific processes, but at the same time respond to the general culture conditions (Dowdy, 2018).”
-------
(Comment 2.7) “in samples taken between 15 to 20 days after inoculation under aeration conditions, some race A colonies contained cells with a different morphology having an increased cell size in comparison to most of the pear-shape normal cells (Figure 2A) “ – it would be good to indicate those cells in the colony image (panel A) as it is not clear which cells are larger… generally, it is hard to differentiate individual cells in panel A (cells seem almost fused/overlapping; is it a multilayered colony?). What are the blue spots in panel B? Also, in Figure 2 in addition to stating that the samples were taken from cultures after 15 days of growing it would be good to know the population density as that would likely influence cell physiology and cell cycle progression; and if the two strains were at different cell densities (and different points in the population growth curve) they might show different responses just due to such differences (not due to strain-specific differences)
(Answer 2.7) With the purpose to make a clear indication of the different cells in race A colonies, new images are included as Fig. 2, 3, 4 and 5 to show more clearly the larger and multinucleated cells without chlorophyll. We hope they are more acceptable for the reviewers. As mentioned before, colonies contain about 50 to 100 cell per colony and cells are arranged at random like in a bunch of grapes, not in an ordered multilayer. So, it is difficult to focus most cells properly. Blue spots in some pictures are from DAPI staining because samples were double stained with SYBR-Green-DAPI. Unfortunately, we do not have the population density of those samples. We only know that the OD 680 nm at 15 days was 0.162 ± 0.018 for race A and 0.150 ± 0.003 for race B, according to the growth curve of each race. Samples were at mid-log phase when cultures were at maximal metabolic activity.
-------
(Comment 2.8) “To check if these differences were maintained along the culture cycle, the aeration of the B. braunii A and B race cultures was stopped when the cultures reached the stationary phase after three weeks of growth. Then, the flasks were kept in this non-aerated condition with the same photoperiod for three more weeks. “. Again – if the question was to investigate changes along the culture cycle, why stopping the aeration? Why including a new factor? Is this part of the culture cycle for hydrocarbon production???
(Answer 2.8) As mentioned before, aeration was stopped to reduce the metabolic activity with the intention of synchronizing the cells in the culture, which has been the main problem we have to study the cell cycle of this alga. Mononucleated race A cells were observed in samples took after 21 days under this non-aerated condition to check synchronization. Unfortunately, we are still unable to get this synchronization, and we agree with the reviewer that many other factors affect cell biology in the stationary phase. Up to now, we do not have sufficient experimental evidence to assume that lack of aeration was the reason of the presence of mononucleated cells. Because of this reason, we decided to eliminate all data regarding non-aerated conditions. However, our data support the presence of multinucleated cells without chlorophyll present just in race A during aerated and active metabolic conditions after 15 days of inoculation. It is known that hydrocarbon production occurs along the whole cell cycle mainly when cells are dividing (Hirose et. al., 2013).
-------
(Comment 2.9) “After a further 21 days in the non-aerated condition, samples were carefully taken from the colonies on the surface. “. Why only from the surface? These colonies have likely experienced less the “non-aerated conditions” since they are at the medium-air interface….
(Answer 2.9) We agree with the reviewer, cells at the surface are not really under non-aerated conditions. Colonies will float since the hydrocarbons continually accumulate. However, at that stage, growth conditions were really unknown because nutrients were seriously depleted and the empty space over the floating cells may contain oxygen and volatiles from stressed cells. These and other factors will affect the cell cycle and physiology of the colonies and individual cells. Because of these reasons and as mentioned before, we decided to eliminate all data regarding non-aerated conditions.
-------
(Comment 2.10) Figure 3: the nuclei in panel A seem a lot smaller than the nuclei in panel B… Is it possible that those are in facts nuclear fragments? Maybe as part of the activation of a stress-induced programmed cell death process? That would also explain the lack of chlorophyll/fluorescence… Mitochondria tend to change their morphology during stress/PCD too… Stress is also consistent with the presence of those red-orange cells in Figure 4D that the authors suggest are the result of stress…
(Answer 2.10) Some measurements we made indicated that in race A mononucleated cells the nuclei are 2.56 ± 0.25 mm, and in race A multinucleated cells (Fig. 3A) the nuclear size was smaller (1.40 ± 0.14 µm). In race B mononucleated cells (Fig. 3B), the nuclear size was 1.91 ± 0.21 µm. We agree that blue spots could be nuclear fragments associated to induced programmed cell death (PCD) mechanism, which also may correlate with the lack of chlorophyll as described in C. reinhardtii (Liu et. al., 2021). Also, changes in the mitochondrial morphology have been reported during PCD for C. reinhardtii and A. thaliana (Scott & Logan et al., 2008). We also agree that the red pigments we observed in Fig. 4D may be carotenoids present during some stress conditions, but as we mentioned in the manuscript, other pigments may be detected by fluorescence like pheophytins (550 – 600 nm), a degradation product of chlorophyll when it loses the Mg2+ ion (Giampaoli et. al.,2020). However, we did not induce PCD intentionally and samples showing multinucleated cells without chlorophyll were taken during the active metabolic stage at 15 days after inoculation when no stress was expected as far as we know. On the other hand, we did not observe any nuclei fragmentation or lack of chlorophyll in any race B samples taken at same times. So, it is hard to think that just race A was under some stress condition able to result in PCD but not race B at same metabolically active stages.
-------
(Comment 2.11) In Figure 4 panel A - are all cells multinucleate? And in panel D, again, the nuclei in “multinucleated cells” appear smaller than the nuclei in mononucleated cells … Where the light-green cells assessed for viability? They look dead or autophagic... Any suggestions as to why only some cells in the colony show this phenotype?
(Answer 2.11) In Figure 4A most cells are multinucleated, but not all. It is hard to distinguish mononucleated cells because this sample was stained just with DAPI. As we mentioned in the manuscript, DAPI also stains other structures such as polyphosphates, nuclei, and nucleoids (Vítová et. al., 2005; Khanzada 2020). Figure 4A was included to show nuclei and nucleoids of race A cells in blue and several poly-P in a yellow-green fluorescence (white arrows). Interaction of poly-P with DAPI shifts the emission maxima from 456 nm for DNA to 526 nm for poly-P (Mukherjee & Ray, 2015). Thus, nuclear DNA and poly-P stained by DAPI can be distinguished by emission detection. In the Discussion, we propose that putative nucleoids could be seen in both B. braunii races, but race A seems to have more nucleoids and poly-P than race B (Figs. 4A and 4B). However, in Figure 4D if the blue DAPI stained spots are nucleoids attaching the dye to the minor grove of A-T base pairs (Kapuscinski, 1995), SYBR Green staining should make them greenish (Skeidsvoll & Ueland, 1995; Marie et. al.,1997) if DNA is present. It is known that DAPI can also bind to RNA (emitting light at c. 500 nm) and to Pi‐rich compounds, including inositol polyphosphates and poly-P (emitting light at c. 550 nm). Thus, DAPI cannot distinguish between inorganic and inositol polyphosphate molecules (Lorenzo‐Orts et. al., 2020). So, in our case the blue DAPI staining shown in Figure 4D may be due to other kinds of poly-P polymers, perhaps not so dense or large as those of yellow-green color. Another possibility is that they are a kind of adenine and thymine storage or other kind of poly-P or RNA having the emission maxima around 456 nm. Regarding the size of the nuclei, yes the nuclei in “multinucleated cells” is smaller than the nuclei in mononucleated cells as described before. The light-green cells were not assessed for viability, and as mentioned before, we did not observe any signal of dead or autophagy in race B samples taken at same times. Again, it is hard to think that just race A was under some stress condition able to result in specific damage but not race B; however, we can't discard this possibility. We suggest that this phenotype of multinucleated and lack of chlorophyll cells is because they are under multiple fission process. We do not know why just some cells are showing this phenotype and not others inside the same colony. However, if they were dead cells or in the process of dying at some point we should see all those cells brownish and wrinkled into the colony and perhaps isolated single cells with this phenotype, but we did not see any of these characteristics in our observations.
-------
(Comment 2.12) “Many green algae such as carteri and C. reinhardtii are known to use palintomy, or multiple fission to proliferate [109]. In this context, our results suggest that B. braunii race A may have cells that undergo multiple fission in specific culture conditions. “. The two algae mentioned above do undergo multiple fission, but they are never multinucleated; cytokinesis immediately follows every mitotic event. So, they are not relevant to what the authors try to suggest here….
(Answer 2.12) The reviewer is correct. So, we modified the sentence as follows to be more relevant for the purpose of our manuscript: “Many green algae such as Scenedesmus sp. and Chromochloris sp. are known to use palintomy, or multiple fission, to proliferate. Multinucleated cells are present in both algae were nuclei in mononucleated cells are bigger that nuclei of multinucleated cells (Koren et. al., 2021). In this context, our results are similar and suggest that B. braunii race A colonies may have cells that undergo multiple fission in multinucleated cells”.
-------
(Comment 2.13) “If multiple fission occurs in race A grown under the aerated and metabolically active condition, this situation may be similar to that in some Volvocales that were grown under illuminated periods that allows for metabolically active conditions [99, 42]. “. Not sure what is meant here, as Volvocales always divide via multiple fission…
(Answer 2.13) Attending to this observation, the sentence was modified with the intention to make it clearer as follows. “Multiple fission of Volvocales is favored under metabolically active growing conditions such as optimal light, nutrients, and temperature (Herron 2016; Ivanov et. al., 2016), and in our case race A show multinucleated cells and putative multiple fission when growing at optimal conditions 15 days after inoculation.”
-------
(Comment 2.14) In several places, including Figure 2, SYBR is misspelled (“SYBER”)
(Answer 2.14) We apologize for this mistake. All SYBER words were corrected to SYBR.
Reviewer 3 Report
The manuscript from Cruz et al. describes in silico identification of cell cycle genes such as RBR and CDKs in Botryococcus braunii, a useful microalga producing high amounts of hydrocarbons, which can be processed into fuels. In addition, they found occurrence of multinucleate cells without chlorophyll in a specific race of this microalga, suggesting difference of cell division process between different races. These findings and observations are of certain value as a basic information for further studying this microalga. However, there are some critical issues that should be resolved, especially on the poor resolution of the microscopic observation when describing the subcellular structures and discussing the cell division process in difference races.
Lines 250-252, page 6
Genes homologous to RBR and CDKs were searched in transcriptome sequence of race A and genomic sequence of race B. I wonder whether these sources of sequences really cover the entire genomes of these races. The authors concluded that CDKC has not been found in race A and CDKH2 is absent in race B. However, if the transcriptomic and genomic sequences used here are not completely covering the whole genome, failure in finding of CDK homologs does not mean the absence of specific type of CDKs in these races. If this is the case, difference in the number of CDK genes between the two races does not necessarily reflect their actual difference, and thus should not be used as an explanation for the different cell division processes between these races.
Line 258, page 6
LXCXE motif should be rephrased as LXCXE motif-binding site.
Phosphorylation of RBR is surely important, but difference in phosphorylation sites and their number may not tell much for the function of RBR in different species or races. If similar comparison is made for flowering plants, such as Arabidopsis, rice, soybean, tomato, and so on, there should be some differences in phosphorylation sites of RBR. But such differences do not necessarily indicate the different RBR functions. Without descent analysis of actual phosphorylation at the potential sites and their functional evaluation, the number of potential phosphorylation sites itself barely means anything.
Lines 293-296, page 7
The authors describe that CDKs found in race A belong to nine classes. However, if my understanding is correct, they should belong to 7 (A, B, D, G, H, I and E) instead of 9 classes.
Lines 310-312, page 8
The description related to CDKG is confusing for me. In my understanding, CDKG represents a group of CDKs that share certain common amino acid sequences and show high sequence similarity to each other. However, in the manuscript, it is written that some CDKGs constitute a clade in phylogenetic analysis, but others belong to a completely different clade and show higher similarity to CDKE. Why CDKs present in far different clade are classified in the same CDK group, CDKG? Further explanation may be required for resolving this issue, since readers of this paper should be similarly confused.
Figure 2
The authors described that there are some enlarged and multinucleate cells that do not contain chlorophyll in race A. However, images in Fig. 2A-D do not clearly show what is described in the text. Especially, cell outlines cannot be seen clearly in bright field image (Fig. 2A), making it impossible to evaluate if multiple nuclei (shown in yellow circles) are contained in single cells. Higher magnification and better resolution would be required. In this sense, bright field image in Fig. 4C seems to be much better for showing the presence of cells without chlorophyll.
Figure 3
In the text, the authors write “outline of the chloroplast”, but it is not clear what this term means and how it looks like in the image of Fig. 3. The authors describes that there are multinucleate cells in Fig. 3A, but, again, the cell outlines are not clearly seen in the images in Fig. 3A, making it unable to recognize multinucleate cells. Nuclei of race A (Fig. 3A) seems to be much smaller than those in race B (Fig. 3B). However, in Fig. 2, such clear difference in nuclear size cannot be seen. Further explanation is needed to clarify this issue.
Figure 3
In Fig. 3B, many of nuclei seem to be located close to other nuclei, making a pair of nuclei. I wonder why they are making pairs? Does this reflect frequent cell division of race B?
Figure 4
Based on only simple microscopic observations of bright field and epifluorescence, the authors determined subcellular structures such as coenobium-like structure, nucleoids, and polyphosphate bodies. For example, I do not understand what is pointed by white arrows in Fig. 4A and 4B, and thus why it is possible to determine these fluorescent signals as polyphosphate bodies? For proper determination of these subcellular structures, more detailed observation such as transmission electron microscope (TEM) observation seems to be additionally required. Without such decent analysis with a sufficient resolution, I believe that Figure 4 in the present form does not provide reasonable information to decide which signals correspond to which subcellular structures.
Author Response
Reviewer 3
(Comments)
The manuscript from Cruz et al. describes in silico identification of cell cycle genes such as RBR and CDKs in Botryococcus braunii, a useful microalga producing high amounts of hydrocarbons, which can be processed into fuels. In addition, they found occurrence of multinucleate cells without chlorophyll in a specific race of this microalga, suggesting difference of cell division process between different races. These findings and observations are of certain value as a basic information for further studying this microalga. However, there are some critical issues that should be resolved, especially on the poor resolution of the microscopic observation when describing the subcellular structures and discussing the cell division process in difference races.
Lines 250-252, page 6
(Comment 3.1) Genes homologous to RBR and CDKs were searched in transcriptome sequence of race A and genomic sequence of race B. I wonder whether these sources of sequences really cover the entire genomes of these races. The authors concluded that CDKC has not been found in race A and CDKH2 is absent in race B. However, if the transcriptomic and genomic sequences used here are not completely covering the whole genome, failure in finding of CDK homologs does not mean the absence of specific type of CDKs in these races. If this is the case, difference in the number of CDK genes between the two races does not necessarily reflect their actual difference, and thus should not be used as an explanation for the different cell division processes between these races.
(Answer 3.1) To give a better description to the reviewer on this matter we can add that the source of genome sequences for B. braunii race B (Browne et. al., 2017), the entire genome, consists of 184,385,342 bp in 2,752 scaffolds (N50 = 373 kb) with 49.6% of G + C content and 1,148 holes (4,611 Mb). 18,726 genes have been detected which provides a solid basis for functional and comparative analyzes that facilitates elucidation of the genetic basis for metabolism in this alga (Browne et. al., 2017). The size of the genome of the B. braunii race A (Yamanka strain) is 166.0 ± 0.4 Mb (mean ± SE; n = 2) (Weiss et. al., 2011). However, so far, the genome of this race has not been sequenced. As mentioned in the document, the Devarenne lab contains the transcriptome database for B. braunii race A (Yamanaka strain) (personal communication). The transcriptome is known to be the complete set of transcripts in a cell for a specific developmental stage or physiological condition (Wang et. al., 2009). Therefore, there is a possibility that sequences are missing and more genes could still be identified. The observation of the reviewer is correct regarding the absence of CDKH2 in race B, and considering the Comment 3.3 about RBR role, we will modify the paragraph (lines 494 to 505) as described in the Answer 3.3.
-------
Line 258, page 6
(Comment 3.2) LXCXE motif should be rephrased as LXCXE motif-binding site.
(Answer 3.2) The reviewer is correct, LXCXE has been replaced with LXCXE motif-binding site.
-------
(Comment 3.3) Phosphorylation of RBR is surely important, but difference in phosphorylation sites and their number may not tell much for the function of RBR in different species or races. If similar comparison is made for flowering plants, such as Arabidopsis, rice, soybean, tomato, and so on, there should be some differences in phosphorylation sites of RBR. But such differences do not necessarily indicate the different RBR functions. Without descent analysis of actual phosphorylation at the potential sites and their functional evaluation, the number of potential phosphorylation sites itself barely means anything.
(Answer 3.3) We agree with the reviewer about the necessity to experimentally assay the phosphorylation sites of the RBR protein and their functional importance. We are currently looking for the best way to do this in these two B. braunii races. However, according to the literature phosphorylation levels of RBR proteins are very important for their function in different phases or transitions of the cell cycle. In plants, there are some reports proposing that increased phosphorylation states of the PsRBR1 protein have an important role in the regulation of E2F-mediated gene expression involved in the G1/S transition and DNA replication during the dormancy-to-growth transition in pea axillary buds (Shimizu-Sato 2008). Also, non-cell cycle functions of RBR proteins are also found in the colonial alga Volvox carteri where the RBR1 gene has a gender-specific expression pattern in female cells. The presence of 4 splice variants and 15 potential cyclin-dependent kinase phosphorylation sites suggests that VcRBR1 is subject to control at the posttranscriptional and posttranslational levels. (Kianianmomeni et al, 2008). Given these studies, and with the intention to avoid misinterpretations regarding the putative role of the phosphorylation sites of B. braunii, we propose to modify the paragraph on lines 494 to 505 as follows: “These results propose differences in the potential phosphorylation sites of RBR, in the CDK types, and hence in the respective variations of PSTAIRE motifs within the B. braunii race A and race B CDKs, which could end in differences between the cell cycles of these two races. It is known that different RBR phosphorylation states may control different and specific processes, but at the same time respond to the general culture conditions (Dowdy, 2018). This could be the case of metabolically active B. braunii race A cultures, where most cells seem to be in different stages of the cell cycle process and the race A RBR forms could contain up to 23 different mP-RBR proteins, three more than in race B (Table 1). It has been reported that the phosphorylation state of pea RBR, has a role in the regulation of E2F-mediated gene expression involved in the G1/S transition and DNA replication during the dormancy-to-growth transition (Shimizu- Sato 2008). Also, RBR may have a novel role to specify the Volvox carteri gender (Kianianmomeni et al, 2008). Race A seems to have one additional CDKE2 compared to race B, as well as one putative CDKH2, not found in any other organism studied in this work (Table 2). These putative proposed enzymes could have a role in the cell cycle of these races. However, further experimental analyses must be conducted to determine the function of the putative proteins”.
-------
Lines 293-296, page 7
(Comment 3.4) The authors describe that CDKs found in race A belong to nine classes. However, if my understanding is correct, they should belong to 7 (A, B, D, G, H, I and E) instead of 9 classes.
(Answer 3.4) We appreciate the observation. Indeed, we made an error in the description. We have corrected as follows: “The CDKs found in race A belong to 7 classes (A, B, D, E, G, H, and I), of which two sequences CDKE (CDKE1, CDKE2) and CDKH (CDKH1, CDKH2) were identified in race A”.
-------
Lines 310-312, page 8
(Comment 3.5) The description related to CDKG is confusing for me. In my understanding, CDKG represents a group of CDKs that share certain common amino acid sequences and show high sequence similarity to each other. However, in the manuscript, it is written that some CDKGs constitute a clade in phylogenetic analysis, but others belong to a completely different clade and show higher similarity to CDKE. Why CDKs present in far different clade are classified in the same CDK group, CDKG? Further explanation may be required for resolving this issue, since readers of this paper should be similarly confused.
(Answer 3.5) The reviewer is correct, CDKG represents a group of CDKs that share certain common amino acid sequences and show high sequence similarity to each other. As mentioned in the manuscript, the sequence of these proteins around the PSTAIRE motif or variations on this motif were aligned. We initially did a Neighbor-joining analyses and original Fig. 1 show that some CDKGs constitute a clade in phylogenetic analysis, but others belong to a completely different clade with higher similarity to CDKE. After recommendation from reviewer 2, a Maximum-likelihood analysis was carried out. Because some of the inferred relationships were unexpected, bootstrap values were low and Neighbor- joining analyses were not ideal. The new Fig. 1 (below) shows that the CDKGs are now grouped in three clades (see below). Those from both B. braunii races, O. tauri and A. thaliana are close, but separated from those of C. reinhardtii, G. pectoral, and V. carteri f. nagariensis. In the Supplementary Figure S3, it can be seen a (V)ALK(K) sequence quite similar among the B. braunii CDKG1s of both races, O. tauri CDKG1 and CDKG2, and A. thaliana CDKG;1 and CDKG;2. This sequence is different from the corresponding LAIK(K) sequence shared by C. reinhardtii CDKG1 and CDKG2, G. pectorale CDKG1, and V. carteri f. nagariensis CDKG1. These and other sequences with similar characteristics besides the PSTAIRE motif, may be the reason of this separation. It is possible that these CDKGs evolved differently (diverged), with those of the Volvocales going in one direction and the rest in other direction. So, they are grouped in different clades perhaps because they fulfill similar functions to the most related group.
-------
Figure 2
(Comment 3.6) The authors described that there are some enlarged and multinucleate cells that do not contain chlorophyll in race A. However, images in Fig. 2A-D do not clearly show what is described in the text. Especially, cell outlines cannot be seen clearly in bright field image (Fig. 2A), making it impossible to evaluate if multiple nuclei (shown in yellow circles) are contained in single cells. Higher magnification and better resolution would be required. In this sense, bright field image in Fig. 4C seems to be much better for showing the presence of cells without chlorophyll.
(Answer 3.6) We appreciate the observation, and we include new Fig. 2, 3, 4 and 5 (below) to show more clearly the larger non-chlorophyll and multinucleated cells. We hope they are more acceptable for the reviewers.
-------
Figure 3
(Comment 3.7) In the text, the authors write “outline of the chloroplast”, but it is not clear what this term means and how it looks like in the image of Fig. 3. The authors describes that there are multinucleate cells in Fig. 3A, but, again, the cell outlines are not clearly seen in the images in Fig. 3A, making it unable to recognize multinucleate cells. Nuclei of race A (Fig. 3A) seems to be much smaller than those in race B (Fig. 3B). However, in Fig. 2, such clear difference in nuclear size cannot be seen. Further explanation is needed to clarify this issue.
(Answer 3.7) By “outline of the chloroplast” we mean that the mitochondrial net in B. braunii resembles the morphology of the chloroplast of B. braunii, which is cup-shaped (Suzuki et al., 2013). With respect to old Figure 3 which is replaced by a new Fig. 6 (below), cells having multiple nuclei have been enclosed with a yellow line. In race A mononucleated cells, the nuclear size is 2.56 ± 0.25 µm and in race A multinucleated cells the nuclear size is 1.40 ± 0.14 µm. In mononuclear cells of race B, the nuclear size is 1.91 ± 0.21 µm. Indeed, the nuclear size of multinucleated cells of race A appears to be smaller than the nuclei of race B.
-------
(Comment 3.8) In Fig. 3B, many of nuclei seem to be located close to other nuclei, making a pair of nuclei. I wonder why they are making pairs? Does this reflect frequent cell division of race B?
(Answer 3.8) The new Fig. 6B shows some cells of race B in the process of cell division (binary fission), therefore pairs of nuclei are observed. Yes, it reflects the type of cell division characteristic of race B.
-------
Figure 4
(Comment 3.9) Based on only simple microscopic observations of bright field and epifluorescence, the authors determined subcellular structures such as coenobium-like structure, nucleoids, and polyphosphate bodies. For example, I do not understand what is pointed by white arrows in Fig. 4A and 4B, and thus why it is possible to determine these fluorescent signals as polyphosphate bodies? For proper determination of these subcellular structures, more detailed observation such as transmission electron microscope (TEM) observation seems to be additionally required. Without such decent analysis with a sufficient resolution, I believe that Figure 4 in the present form does not provide reasonable information to decide which signals correspond to which subcellular structures.
(Answer 3.9) We appreciate the comments of the reviewer, and we would like to mention that specifically a coenobium is considered a colony containing a fixed number of elements, with little or no specialization. They occur in several groups of algae. They are often embedded in a mucilaginous matrix and may be motile or non-motile. Examples include Volvox and its relatives, Scenedesmus, Pediastrum, and Hydrodictyon. According to this definition, we detected coenobium-like structures as shown in the amplified right panel of new Fig. 2B.
White arrows in Fig.4, indicate yellowish polyphosphate granules in B. braunii cells. As described in section 3.6. Other DAPI stained structures, DAPI also stains other structures such as polyphosphates (Vítová et. al., 2005; Khanzada, 2020), this fluorescent marker interacts with poly-P in high concentrations to form DAPI-poly-P complexes that allow poly-P rich granules to be visualized by fluorescence microscopy at an emission wavelength of 475–525 nm, which produces a yellowish color. This DAPI staining method is commonly used to detect poly-P-rich organelles in vivo and in vitro (Vítová et al., 2005, Aschar-Sobbi et al., 2008, Ota et al., 2016, Moudříková et al., 2021). More specifically and as described in the manuscript, SYBR-Green I stain DNA in green (it is maximally excited at 497 nm, but also has secondary excitation peaks at ~290 nm and ~380 nm), and DAPI stains DNA in blue (DAPI- DNA complex, Ex = 364 nm, Em = 454 nm) and ortho-polyphosphate (poly-P) in yellow-green (Kapuscinski, 1995; Vítová et. al., 2005). Also, Vítová et. al., (2005) reported that DAPI may stain S. quadricauda nuclei and nucleoids in blue and poly-P in yellow-green. This is because the interaction of poly-P with DAPI shifts the emission maxima from 456 nm for DNA to 526 nm for poly-P (Mukherjee and Ray, 2005). Thus, nuclear DNA and poly-P stained by DAPI can be distinguished by emission detection. We agree with the reviewer that the resolution of TEM (0.1 nm and lower) is higher than the resolution of light microscopy (∼300 nm). However, according to Christ et. al., (2020), and after comparison of 18 different methods to analyze poly-P, they concluded that there is not a significant difference between TEM and fluorescent light microscopy. Briefly, prior poly-P extraction is not required in any of these procedures, both have medium specificity for poly-P and can quantitate total poly-P as well as the cellular location. TEM can detect poly-P of less than 15 P-subunits, and the cation composition, but not fluorescent light microscopy. However, fluorescent light microscopy is more economical and sample preparation is simpler. In TEM, the poly-P specificity can be increased by some techniques such as lead - sulfide staining, electron spectroscopic imaging, or energy-filtered TEM, which allows the detection of poly-P as P. Furthermore, TEM can be coupled with energy dispersive X-ray spectroscopy to detect the inorganic counterions of poly-P, but all these procedures are more expensive and require technical expertise we do not have access to. Finally, it is true that TEM of multinucleated and mononucleated cells should be approached to confirm the differences, and it is our intention to do this in the near future. However, for this initial approach we consider that DAPI stain is sufficient to confirm that poly-P is present in both races and SYBR-Green distinguished the nuclei from other DAPI- stained components. In addition, and as mentioned in the manuscript, poly-P has already been described in B. braunii using TEM (Wolf and Cox, 1981).
With the intention to show clearer and specifically the poly-P in our work, we substituted the Fig. 4 for a new Fig. 7 (below) where putative nuclei and nucleoids could be seen in blue in both B. braunii races (Fig. 7A, 7B), and poly-P are in yellowish color in Fig. 7C. Race A has more poly-P than race B.